# Contemporaneous Inflammatory, Angiogenic, Fibrogenic, and Angiostatic Cytokine Profiles of the Time-to-Tumor Development by Cancer Cells to Orchestrate Tumor Neovascularization, Progression, and Metastasis

**DOI:** 10.3390/cells13201739

**Published:** 2024-10-20

**Authors:** Elizabeth Skapinker, Emilyn B. Aucoin, Haley L. Kombargi, Abdulrahman M. Yaish, Yunfan Li, Leili Baghaie, Myron R. Szewczuk

**Affiliations:** 1Faculty of Arts and Science, Queen’s University, Kingston, ON K7L 3N9, Canada; 21ess18@queensu.ca (E.S.); 18yl210@queensu.ca (Y.L.); 2Faculty of Science, Biology (Biomedical Science), York University, Toronto, ON M3J 1P3, Canada; emilynaucoin@gmail.com; 3Faculty of Health Sciences, Queen’s University, Kingston, ON K7L 3N9, Canada; 20hlk@queensu.ca (H.L.K.); a.yaish@queensu.ca (A.M.Y.); 4Department of Biomedical and Molecular Sciences, Queen’s University, Kingston, ON K7L 3N9, Canada; 16lbn1@queensu.ca

**Keywords:** inflammatory, angiogenic, fibrogenic, and angiostatic cytokines, pancreatic cancer cells, tumor development

## Abstract

**Simple Summary:**

Stromal cells and growth factors play important roles during tumor initiation and progression. Growth factors not only mediate normal biological processes such as development and tissue repair but also tumorigenesis by contributing to tumor growth and the transformation of neoplastic cells. This study investigated the host angiogenic and proinflammatory cytokines during tumor initiation and progression in heterotopic xenografts of MiaPaCa-2-eGFP tumors growing in RAGxCγ double mutant mice. Tumor vasculature requires stringently balanced angiogenic cytokine signaling to provide sufficient vasculature for tumor development. Notably, the mechanism(s) that cause(s) tumors to develop and continuously interact with the tumor microenvironment (TME) and other host systems still need to be investigated. The TME is a complex environment that surrounds and interacts with tumors. The TME generally also includes tumor stroma and immune cells within the tumor node. It is true, though, that tumor cells are embedded in the TME. From a cellular perspective, the TME allows cancerous cells to progress and develop through features such as stromal cells, fibroblasts, endothelial cells, and adipocytes. These components can involve contemporaneous inflammatory and angiogenic cytokine profiles that propitiously prompt tumor survival, local invasion, and metastatic dispersion.

**Abstract:**

Cytokines can promote various cancer processes, such as angiogenesis, epithelial to mesenchymal transition (EMT), invasion, and tumor progression, and maintain cancer stem-cell-like (CSCs) cells. The mechanism(s) that continuously promote(s) tumors to progress in the TME still need(s) to be investigated. The data in the present study analyzed the inflammatory, angiogenic, fibrogenic, and angiostatic cytokine profiles in the host serum during tumor development in a mouse model of human pancreatic cancer. Pancreatic MiaPaCa-2-eGFP cancer cells were subcutaneously implanted in RAG2xCγ double mutant mice. Blood samples were collected before cancer cell implantation and every week until the end point of the study. The extracted serum from the blood of each mouse at different time points during tumor development was analyzed using a Bio-Plex microarray analysis and a Bio-Plex 200 system for proinflammatory (IL-1β, IL-10, IFN-γ, and TNF-α) and angiogenic and fibrogenic (IL-15, IL-18, basic FGF, LIF, M-CSF, MIG, MIP-2, PDGF-BB, and VEGF) cytokines. Here, we find that during cancer cell colonization for tumor development, host angiogenic, fibrogenic, and proinflammatory cytokine profiling in the tumor-bearing mice has been shown to significantly reduce host angiostatic and proinflammatory cytokines that restrain tumor development and increase those for tumor growth. The proinflammatory cytokines IL-15, IL-18, and IL-1β profiles reveal a significant host serum increase after day 35 when the tumor began to progress in growth. In contrast, the angiostatic cytokine profiles of TNFα, MIG, M-CSF, IL-10, and IFNγ in the host serum revealed a dramatic and significant decrease after day 5 post-implantation of cancer cells. OP treatment of tumor-bearing mice on day 35 maintained high levels of angiostatic and fibrogenic cytokines. The data suggest an entirely new regulation by cancer cells for tumor development. The findings identify for the first time how pancreatic cancer cells use host cytokine profiling to orchestrate the initiation of tumor development.

## 1. Introduction

Time-to-tumor progression and growth to metastatic disease is highly dependent on the formation of tumor vascularization, the process of which is called angiogenesis. The impact of angiogenesis on disease progression in solid tumors has been demonstrated in studies on preclinical [1,2] as well as clinical data showing the effect of anti-angiogenic treatment in patients with colorectal cancer [3,4] and non-small-cell lung cancer [5]. Pancreatic tumors do not have extensive vascularization, and thus, the role of angiogenesis for tumor progression in patients with pancreatic cancer is a complex process. For example, Annese et al. [6] reported that pancreatic cancer angiogenesis can be activated by genetic and epigenetic alterations as well as by cellular stromal components of the tumor microenvironment. They proposed that transcription factors should be considered for the development of new anti-tumor and anti-angiogenic therapeutic approaches. Grobbelaar et al. [7] reported on the role of angiogenesis regulators promoting disease progression in pancreatic cancer and how these molecules impact resistance to gemcitabine and various therapies against pancreatic cancer. Therefore, research is still needed before a gold treatment standard can be made [8]. Understanding angiogenic drug resistance also highlights the important need for additional research to regulate alternative non-VEGF-related pro-angiogenic pathways. For example, the potential efficacy of anti-angiogenic therapy for pancreatic cancer [9,10], including controlled clinical trials on anti-VEGF (vascular endothelial growth factor) monoclonal antibody bevacizumab and anti-EGFR (epidermal growth factor receptor) monoclonal antibody cetuximab, failed to demonstrate a survival benefit of these anti-angiogenic therapies for patients [11,12]. The failure of anti-angiogenic therapeutic clinic trials for pancreatic cancer may result from targeting one molecule or its receptor. Studies on the mechanism underlying angiogenesis and the prognostic outcome of angiogenic cytokines in pancreatic cancer have been limited to a single or a few molecules.

Angiogenesis is a critical process in the development and progression of tumors. It involves the formation of new blood vessels from pre-existing capillaries or post-capillary veins [13]. Under normal physiological conditions, the growth and maturation of these new vessels are tightly regulated [13]. The process of angiogenesis relies on a delicate balance between promoters and inhibitors of blood vessel formation [14]. However, in the context of cancer, this regulatory balance is often disrupted. Tumors manipulate angiogenesis to secure a consistent supply of nutrients and oxygen, thereby facilitating their growth and the potential to spread to other parts of the body [13]. This dissemination leads to the establishment of new cancer cell communities, a phenomenon known as metastasis [15].

Habanjar et al. [16] showed that the innate immune system, including neutrophils and mononuclear macrophage systems, significantly correlates with tumors. The mechanism by which innate immune cells regulate angiogenesis in the tumor microenvironment is that the interaction between cytokines and inflammatory cells is a critical factor in tumor angiogenesis and progression [17]. These cells adopt phenotypes that enhance angiogenesis and switch to tolerant and immunosuppressive behaviors [17]. The release of pro-angiogenic cytokines by tumor-associated macrophages (TAMs) and other inflammatory cells triggers the activation and recruitment of endothelial cells (ECs), which are critical for forming new blood vessels [17]. Chemokines, cytokines, and enzymes facilitate this process [17]. In addition to promoting angiogenesis, TAMs also affect the production of various enzymes and proteases [17]. These enzymes break down ECM components and promote tumor cell invasion and metastasis [17].

Angiogenesis is also closely intertwined with carcinogenesis [18]. Observing the factors that induce angiogenesis is vital to understanding the mechanisms of cancer progression and developing effective treatments. This present study focuses on the relationship of specific cytokines with angiogenesis and tumorigenesis properties, recognizing that while many factors promote angiogenesis, cytokines can play a pivotal role in these processes. For example, angiostatic cytokines are a category of small signaling proteins crucially involved in properly regulating angiogenesis [19]. These cytokines aid in maintaining vascular integrity and prevent the formation of micro-vessels in unwanted areas, such as tumors [19]. For instance, angiostatic cytokines like interleukin-18 (IL-18) bind to specific high-affinity receptors, inhibiting endothelial cell proliferation, migration, and tube formation [19]. This mechanism thus counteracts the effects of pro-angiogenic compounds, like vascular endothelial growth factor (VEGF), which inhibits tumor neovascularization [20]. Tumor angiogenesis can result not only from the interaction of cancer cells with endothelial cells but also with the surrounding inflammatory cells, which have a crucial role in directing the neoformation of blood vessels.

Generally, angiostatic cytokines are produced and secreted by immune cells (e.g., lymphocytes) and can have autocrine, paracrine, or endocrine effects on the body [21]. It is also important to note that some angiostatic cytokines can simultaneously be pro- and anti-inflammatory [19]. While cytokines are not directly obtained from dietary methods, certain nutrients can regulate the production of cytokines [22]. For example, seafood omega-3 fatty acids have anti-inflammatory properties that modulate the production of proinflammatory cytokines [22]. Moreover, cytokines are released at specific times, usually in response to stimuli such as post-operative conditions, infections, and immune disturbances [23].

Notably, Baghaie et al. [24] reported on the clinical significance of the variant profiles of angiogenic and proinflammatory cytokine concentrations in relation to cancer progression post-surgery [24]. The results of the study demonstrated an extreme change in the expression levels of proinflammatory and angiogenic cytokines and molecules [24]. Notably, a significant rapid rise in tumor- and metastases-promoting molecules such as matrix metalloproteinase-9 (MMP-9), interleukin-6 (IL-6), hepatocyte growth factor (HGF), and placental growth factor (PLGF) was observed immediately (2 h) after surgery [24]. Concurrently, there was a significant decrease in transforming growth factor-β1 (TGF-β1), platelet-derived growth factor (PDGF-AB/BB), insulin-like growth factor-1 (IGF-1), and monocyte chemoattractant protein-2 (MCP-2), signifying perioperative immunosuppression [24]. It is important to note that most levels of these molecules returned to normal concentrations within 1–2 weeks post-surgery [24]. In essence, these findings highlight a critical period post-surgery characterized by immunosuppression, enhanced angiogenesis, and tissue remodeling, which collectively promote residual tumor proliferation and metastases [24]. Although most of these cytokines return to baseline levels relatively quickly, this brief inflammatory state may be sufficient in triggering any residual tumor cells to undergo epithelial–mesenchymal transition (EMT) [24]. Additionally, the increase in cytokines such as IL-6 and HGF emphasizes their role in surgery-induced tumor growth and metastasis and the role of other cytokines like MCP-2 in immunosuppression [24]. To conclude, Baghaie et al. [24] suggested that a critical window for therapeutic intervention exists and can be exploited to prevent cancer recurrence.

Cytokines also play a crucial role in regulating epithelial–mesenchymal transition (EMT), a key process in metastasis [25]. EMT in cancer development is marked by the loss of intercellular adhesion structures, changes in cell polarity, and a shift to a spindle-shaped cell morphology [26]. This process involves the downregulation of intercellular connections, a switch from keratin to vimentin intermediate filaments, and enhanced cell invasion and motility [26]. Various cytokines and chemokines produced by tumor cells, cancer-associated fibroblasts (CAFs), and tumor-associated immune cells in the tumor microenvironment (TME) stimulate EMT, thereby promoting cancer metastasis [25]. O’Shea et al. [27] reported the potential of oseltamivir phosphate (Tamiflu^®^), a neuraminidase-1 inhibitor, as an anti-cancer agent that can reverse EMT. This reversal, known as mesenchymal-to-epithelial transition (MET), reduces chemotherapy resistance and enhances the efficacy of existing treatments [27].

Oseltamivir phosphate (OP) has been shown to significantly decrease the activity of chemically resistant pancreatic cancer (PANC1) cells, thus increasing their sensitivity to chemotherapy [27]. Furthermore, OP increases the expression of E-cadherin (an epithelial marker) and decreases the expression of N-cadherin and VE-cadherin (mesenchymal markers) [27,28]. This change in cadherin expression suggests that pancreatic cancer cells treated with oseltamivir phosphate adhere more to surrounding tissues and are less likely to metastasize [27]. N-cadherin promotes cell motility, while VE-cadherin contributes to tumor angiogenesis, and it was observed that a reduction in N-cadherin expression also resulted in decreased VE-cadherin levels [27]. A decline in N-cadherin and VE-cadherin expression leads to reduced cell proliferation and formation of aggressive structures, indicating a reduction in cancer cell aggressiveness [27]. Overall, OP shows promise as an alternative treatment strategy for pancreatic cancer and other cancers exhibiting similar resistance mechanisms by targeting EMT and enhancing the effectiveness of chemotherapy [27].

Here, we investigated the mechanism by which early nascent tumor cells hijack the host immune machinery to facilitate tumor vascularization for tumor progression. Here, we used RAG2xgamma c (Cγ) double mutant mice that are completely alymphoid (T-, B-, NK-cells) and do not form spontaneous tumors but exhibit normal hematopoietic cellular parameters. However, the RAG2xCγ double mutant mice have decreased numbers of dendritic cells, macrophage cells, and neutrophils and lack functional receptors for IL-2, IL-4, IL-7, IL-9, and IL-15 cytokines. Using a highly sensitive Bio-Plex microarray mouse cytokine assay optimized to positive control samples, the time-to-tumor progression revealed an upregulation of pro-angiogenic cytokines and downregulated angiostatic cytokines, ultimately promoting tumorigenesis and metastasis. Biophotonic images of live, non-processed necropsy of tumor, liver, lung, spleen, and heart from the untreated cohort, 2 mg/kg, 10 mg/kg, 50 mg/kg, and 100 mg/kg OP cohorts revealed the extent of visual metastasis. Additionally, OP-treated tumor-bearing animals at the start of tumor development affected the cytokine profiles that upended tumor growth and metastasis.

## 2. Materials and Methods

### 2.1. Cell Lines

Mia-PaCa-2 cells (ATCC^®^ Number: CRL-1420™, Manassas, VA, USA) are human pancreatic cancer cells with floating round cells expressing the beta-estradiol (E2)-binding estrogen receptors and attached epithelial cells. They were derived from a carcinoma exhibited in a 65-year-old white male patient. Cells were grown at 37 °C in 5% CO_2_ in culture media with Dulbecco’s Modified Eagle Medium (DMEM) (Gibco, Rockville, MD, USA) and enriched with 10% fetal calf serum (FCS) (HyClone, Logan, UT, USA).

### 2.2. eGFP Lentifect Purified Lentivirus Particles to Detect Necropsy Tissue Metastasis

GeneCopoeia Lentifect™ Lentiviral Particles (Cat# LP-EGFP. LV105-0205, GeneCopoeia, Inc., Rockville, Maryland 20850, USA) are ready-to-use particles. They are produced from a standardized protocol using purified plasmid DNA and the proprietary reagents EndoFectin™ Lenti (for transfection) and TiterBoost™. The protocol uses a third-generation self-inactivating packaging system meeting BioSafety Level 2 requirements with University Biohazard committee approval. The Lentifect particles include a cytomegalovirus (CMV) promoter for the expression of non-tagged eGFP in target cells, and a puromycin resistance marker is used for the stable selection of transduced cells. The ready-to-use lentiviral particles were used for the transduction of MiaPaCa-2 cells.

Briefly, cells were cultured in 6 well tissue culture plates in DMEM medium containing 10% FCS and 5 μg/mL plasmocin. After 24 h, after the medium was discarded, 2 mL of 5 μg/mL of media containing polybrene was added to the cells, followed by eGFP lentiviral particles at MOI = 6. The plate was mixed and centrifuged at 2500 rpm for 90 min and incubated at 37 °C in a 5% CO_2_ humidified incubator for 24 h. After washing, the cells were re-cultured in media for an additional 2 days. On day 5, the media containing an optimal 2 μg/mL of puromycin, as pre-determined in a cell viability assay, was replaced in the cells. The selection media was added every 40 h to expand puromycin-resistance eGFP-transduced MiaPaCa-2 cell clones. The transfection efficiency of 90% was determined using fluorescence microscopy (Zeiss Imager M2, Carl-Zeiss-Straße 2273447 Oberkochen, Germany) and biophotonic imaging (Cancer Biology & Genetics Division of the Queen’s Cancer Research Institute) before implantation into the mice.

### 2.3. Biophotonic Imaging of MiaPaCa-2-eGFP and Metastasis

Biophotonic images of live freshly non-processed necropsy of tumor, liver, lung, spleen, and heart from the untreated cohort and treated cohorts with 2 mg/kg OP, 10 mg/kg OP, 50 mg/kg OP, and 100 mg/kg OP to reveal the migration of MiaPaCa-2-eGFP to these organs and visualize metastasis. The tissues were placed on a Petri dish, and the images of fresh tissue fluorescence were analyzed with a low-magnification fluorescent microscope. The enhanced green fluorescent protein (eGFP) has an excitation peak at 488 nm (blue light) and emits light maximally at 507 nm.

### 2.4. Oseltamivir Phosphate (OP) Treatment

Oseltamivir phosphate (OP) (La Roche Ltd., Mississauga, ON, Canada) was used at indicated concentrations based on the proof-of-evidence for an effective OP monotherapy in the treatment of human pancreatic cancer growth and metastatic spread in heterotopic xenograft of tumors growing in RAG2xCγ double mutant mice [29]. To test the efficacy of OP at different dosages, mice were intraperitoneally injected with 0.2 mL of OP at 2, 10, 50, 100, and 200 mg/kg in sterile phosphate-buffered saline (pH 7.4) daily or as otherwise indicated starting at day 35 post-implantation.

### 2.5. Cancer Cell Implantation in RAG2xCγ Double Mutant Xenograft Mice

Puromycin-resistance eGFP-transduced MiaPaCa-2 cell clones were grown in a 75 cm^2^ culture flask at 80–90% confluence. The cells were put into solution using TrypLE Express (Gibco) and washed with sterile saline. The cells in solution were centrifuged for 5 min at 900 rpm. The cell pellet was resuspended in sterile normal saline at 6 × 10^6^ cells/mL for 0.2 mL injection of 1.2 × 10^6^ cells subcutaneously into the right back flank of the mouse. At the endpoint of the experiment, mice were euthanized by cervical dislocation, and live necropsy tumors, liver, lung, and heart were biophotonically imaged for fluorescence of MiaPaCa-2-eGFP cells, tumor weights, and hematoxylin and eosin (H&E, Cedarlane Labs, Burlington, Ontario, L7L 5R2, Canada) staining of paraffin-embedded tissues. Humane endpoints of the experimental mice were monitored daily by a registered veterinary practitioner. Mice were euthanized when the tumor volume reached 1000 mm^3^ or a body weight loss of 15%, according to Animal Care Protocol. Tumor measurements were taken twice a week. Tumor volumes were analyzed using the following formula:Volume = (width square) × length × ½

### 2.6. Heterotopic Xenograft Mouse Model of Human Pancreatic Cancer

An immunodeficient mouse model with a double mutation in the combining recombinase activating gene-2 (RAG2) and common cytokine receptor γ chain (Cγ) was used as the xenograft mice. The RAG2xCγ double mutant mice on a BALB/c genetic background are completely alymphoid (T-cell, B-cell, and NK-cell deficient), show no spontaneous tumor formation, and exhibit normal hematopoietic parameters. The mice were made by inter-crossing and were kept in specific pathogen-free isolators in the Animal Care Facility, Queen’s University, Kingston, Ontario K7L 3N6, Canada. SPF animal research facility provides breeding, housing, and procedural space for animals free of a defined list of pathogens. A colony of mice was established in the facility. Mice were cared for under sterile conditions in micro-isolators or air-filtered cages and were given autoclaved food and water. The female mice used in the study were between 6 and 8 weeks of age. The Animal Care Committee approved mice use in the study, protocol number 2017-1708, at Queen’s University.

### 2.7. Mouse Blood Serum Collection

Blood was collected retro-orbitally by penetrating the retro-orbital plexus located behind the mouse’s eye. An amount of 200 µL of blood was collected with a capillary tube. Blood was left to coagulate and then centrifuged. Collected serum was then freshly frozen at −80 °C and analyzed later for cytokine profiling using the Luminex Bio-Plex microarray mouse cytokine assay.

### 2.8. Luminex Bio-Plex Microarray Mouse Cytokine Assay

The Bio-Plex microarray mouse cytokine assay measures the relative amount of the cytokine only compared to optimized positive control mouse samples. Using the multiplex format, the profiles of multiple cytokines in freshly frozen (−80 °C) blood serum were detected using only 50 µL of serum from each mouse. The assay used a 6.5 µm magnetic bead protocol. The assay protocol is optimized for high sensitivity and higher specificity in minimizing cross-reactivity and broad dynamic range using highly specific antibodies for 9 mouse angiogenic cytokines (IL-15, IL-18, basic FGF, LIF, M-CSF, MIG, MIP-2, PDGF-BB, and VEGF) and 8 mouse proinflammatory cytokines (IL-1β, IL-10, IFN-γ, and TNF-α) and adjusted equally for Bio-Plex microarray analyses using the Bio-Plex 200 System. Cross-reactivity against human cytokines was not tested.

### 2.9. Flow Cytometry of Host CD31+ Endothelial Cells in the Blood

Live mouse blood cells were harvested from retro-orbital blood at different times after MiaPaCa-2-eGFP cancer cell implantation subcutaneously. Live blood cells were counted for a final concentration of 1 × 10^6^ cells/mL. All subsequent steps were performed on ice. Cells were washed 2× in 2% fetal bovine serum (FBS) + 1× PBS before the primary antibody addition at 1:300 dilution of a stock solution of 1 mg/mL of a rat monoclonal antibody conjugated with DyLight-488 fluorochrome against mouse CD31/PECAM-1 (Catalog # FAB6874G-100UG, Novus Biologicals Canada ULC, Oakville, ON L6M 2V5 Canada) at a final concentration of 10 μg/mL and incubated for 60 min. The cells were then washed 2× with 2% FBS + 1× PBS and fixed in 1 mL of 4% paraformaldehyde solution before flow cytometry analysis.

Flow cytometry of the presence of the characteristic mouse CD31+ endothelial cells in the host blood was conducted when the MiaPaCa-2-eGFP tumor-bearing RAG2xCγ double mutant mice had received various treatments of oseltamivir phosphate (OP) 50, 100 and 200 mg/kg at day 42.

### 2.10. Hematoxylin and Eosin (H&E) Staining

Fixed tumors and livers were embedded in paraffin, sectioned at 5 µm, and transferred onto glass slides. The sections were deparaffinized through xylene and graded alcohols into the water and stained with hematoxylin and eosin (H&E). The slides were dehydrated and mounted. All stained sections were assessed using a light microscope (Nikon Eclipse SE EI R STG HNDL TRINOC-oil obj set with digital sight 1000 microscope camera, Nikon Instruments Inc., Melville, NY 11747-3064, USA).

### 2.11. Statistical Analysis

Statistical analysis was carried out using GraphPad Prism 5. Results were compared by a one-way ANOVA at 95% confidence using Bonferroni’s multiple comparison test. Asterisks denote *p* < 0.05 as the threshold of significant difference. The log-rank (Mantel–Cox) test was used to test the probability that the survival curves were significantly different using the Chi-square.

## 3. Results

### 3.1. Oseltamivir Phosphate (OP) Impedes the Tumor Growth in the Heterotopic Xenograft of Human Pancreatic MiaPaCa-2-eGFP Cancer Cells in RAG2xCγ Double Mutant Mice

The preclinical in vivo anti-tumor activity of OP was investigated in a RAG2xCγ double mutant xenograft mouse model of human pancreatic cancer. The RAG2xCγ double mutant mice lack mature T cells, B cells, and functional NK cells, leading to better engraftment of human cells than any other mouse strain. Here, MiaPaCa-2-eGFP pancreatic cancer cells were implanted subcutaneously in the right back flank of these mice. Twice a week, following implantation of the cancer cells, each tumor was measured for tumor volume growth (length and width) at the site of implantation, as well as body weight, and body condition scoring. Scoring the body condition of rodents is a non-invasive method for assessing health and establishing endpoints for adults where body weight is not a viable monitoring tool.

The data depicted in Figure 1A clearly demonstrate the effect of daily injection of 100 mg/kg of soluble OP intraperitoneally in sterile saline at day 47 post-implantation when the tumor volume reached > 900 mm^3^ of the untreated control. The data also depicted in Figure 1A show the size of the tumor on the right flank at day 47 post-cancer cell implantation for the untreated and OP-treated tumor-bearing mice. Notably, the tumor at necropsy from the OP-treated cohort showed markedly less neovasculature compared to the untreated tumors. To test the efficacy of OP at different dosages, Figure 1B illustrates the time-to-tumor progression following 2, 10, 50, 100, and 200 mg/kg OP daily injections starting at day 35 post-implantation. Interestingly, the 200 mg/kg OP treatment had a profound diminution of tumor growth compared to the untreated control cohort. There was a significant reduction in tumor wet weight (Figure 1C) from the 50, 100, and 200 mg/kg OP-treated tumor-bearing mice compared to the untreated cohort. This dosing regimen of OP had no side effects as determined by minimum loss of body weight (Figure 1D) and body condition scoring. Figure 1E depicts the survival curves of the OP-treated cohorts, demonstrating a significantly different survival rate (*p* = 0.0115) according to the log-rank (Mantel–Cox) test compared to the untreated cohort.

The data in Figure 2A–C revealed the tumor growth inhibition rate following one, four, and five intraperitoneal injections of OP of 50, 100, and 200 mg/kg at day 42 post-implantation of MiaPaCa-2-eGFP cells. Here, the data demonstrated that OP with five injections inhibited 40–50% of the tumor growth compared to the untreated cohort. Furthermore, we analyzed the host CD31+ endothelial cells in the blood. Live blood cells from each mouse were immunostained for the CD31 marker on the cell surface using flow cytometry. CD31 is also known as PECAM-1 (Platelet Endothelial Cell Adhesion Molecule-1). The CD31 staining results indicated an increase in CD31^+^ blood mouse cells following tumor cell implantation, with a dramatic decline to a flat line at day 20 post-implantation and remaining flat at the time of initial signs of tumor growth. These findings are reproducible in two independent experiments. These data suggest that MiaPaCa-2-eGFP pancreatic cancer cells, following implantation, establish neovascularization for tumor growth and induce the CD31+ mouse’s physiologic system for blood vessel formation and growth (Figure 2D–F).

Figure 2 depicts the necropsy tumor images for (G) untreated, (H) OP 50 mg/kg, and (I) OP 100 mg/kg after eight IP injections. Notably, the tumors from the OP-treated cohorts show fewer bloody vessels than those from the untreated cohort, confirming that OP affects the tumor neovascularization process.

Next, we asked if lower doses of OP would affect tumor neovascularization and prevent cancer cell migration. The data in Figure 3A reveal the biophotonic images of MiaPaCa-2-eGFP tumors and their migration to the liver, lung, spleen, and heart, revealing minimal or negligible metastatic MiaPaCa-2-eGFP of OP-treated cohorts compared to the untreated control. The OP-treated cohorts show minimal tumor neovascularization compared to the untreated cohort (Figure 3B), supporting the biophotonic tumor data in Figure 3A.

### 3.2. Blood Serum Profiles of Proinflammatory, Angiogenic, and Angiostatic Cytokines with Time-to-Tumor Progression

Neo-angiogenesis must occur to carry sufficient nutrients to the tumor for its growth. In addition to growth, the development of new blood vessels is also necessary for invasion and metastases of the tumor. Here, blood serum from each mouse was collected before and every 5 days following subcutaneous implantation of pancreatic MiaPaCa-2-eGFP cancer cells in the right flank. The blood serum was analyzed for cytokine profiles with time-to-tumor progression using a very sensitive Luminex Bio-Plex microarray cytokine assay. The time-to-tumor progression revealed significant host cytokine responses initiated by the cancer cells to establish neovasculature for tumor growth, as depicted in Figure 1A.

Figure 4 shows the proinflammatory (IL-15, IL-18, IL-1β, and TNFα) and angiostatic (MIG, M-CSF, IL-10, TNFα, and IFNγ) cytokine profiles in a time-to-tumor progression over 45 days. Interestingly, IL-15, IL-18, and IL-1β cytokine profiles reveal a significant host serum increase after day 35 when the tumor began to progress in growth (Figure 4A,C,D, respectively). In contrast, the angiostatic cytokine profiles of TNFα, MIG, M-CSF, IL-10, and IFNγ in the host serum revealed a dramatic and significant decrease after day 5 post-implantation of cancer cells.

To explain the cytokine profiles during time-to-tumor growth progression in Figure 4, Yang and Lundqvist [30] reported an excellent review revealing the different roles of IL-2 and IL-15 in modulating T-cells and NK cells under different immune suppressions. Beltra et al. [31] reported that IL-2 and IL-15 both can trigger CD8+ T cell exhaustion by inducing the expression of inhibitory receptors in vivo, particularly 2B4 and TIM-3 and by selectively abrogating their common IL-2R chain and retaining the inhibitory receptor induction. Discerning the role of IL-15 in the regulation of anti-tumor immune responses for immunotherapeutic approaches against cancer, the data in this study suggest the importance of critical timing during tumor development for effective immunotherapy treatments.

Palma et al. [32] provided an excellent review of the multiple roles played by IL-18 in immune regulation, cancer progression, and angiogenesis based on the observation of elevated IL-18 expressions by cancer patients. In particular, the mechanism of action of IL-18 in cancer progression involves hypoxia, which induces the transcription and secretion of IL-18, followed by the expression of hypoxia-inducible factor-1α (HIF-1α) [33]. Here, HIF-1α mediates tumor progression. For IL-1β, Rébé and Ghiringhelli [34] described that this cytokine has pleiotropic effects on immune cells, angiogenesis, cancer cell proliferation, migration, and metastasis. However, in contrast, any anti-cancer treatments are able to promote IL-1β production by cancer or immune cells, with opposite effects on cancer progression. Similarly, TNFα has properties to stimulate proliferation, survival, migration, and angiogenesis in most cancer cells that are resistant to TNF-induced cytotoxicity, resulting in tumor promotion [35]. Interestingly, in Figure 4G, the data reveal high TNFα serum levels at an early stage of tumor development on day 5, with a progressive significant decline after day 10, until the tumor begins to grow, on day 35.

In addition, we have reported on the effectiveness of oseltamivir phosphate (OP) monotherapy on pancreatic cancer tumor growth and metastatic spread in the heterotopic xenograft of tumors growing in RAG2xCγ double mutant mice [29]. Taken together, the findings in the report identified, for the first time, that neuraminidase-1 (Neu1) is an important cancer-targeting enzyme that is unaffected by the activating mutations in cancer cells. In addition, Neu1 was shown to play a central role in mediating Toll-like (TLR) activation and signaling [11,36]. The signaling interactions involving neuromedin B GPCR (NMBR)–MMP9–Neu1 crosstalk constitute a novel cell surface and intracellular TLR signaling platforms that are essential for NF-κB activation and immune proinflammatory responses.

### 3.3. Dose-Dependent Effect of Oseltamivir Phosphate (OP) on Angiogenic IL-15, IL-18, and M-CSF Cytokine Profiles Affecting Pancreatic MiaPaCa-2-eGFP Tumor Growth in Heterotopic Xenograft Mice

We have reported that OP impeded tumor neovascularization, growth, and metastasis of human triple-negative breast [37] and pancreatic [29] cancer cells in heterotopic xenografts of these tumors in RAG2xCγ double mutant mice. Hrynyk et al. [38] reported that a metronomic therapy of slow-release OP from poly (lactic-co-glycolic acid) (PLGA)-loaded OP impeded tumor vascularization, growth, and invasiveness in heterotopic xenografts of PANC-1 tumors in RAG2xCγ double mutant mice. Here, we investigated the dose-dependent effect of OP on angiogenic cytokine profiles in a time-to-tumor progression in heterotopic pancreatic MiaPaCa-2-eGFP xenograft mice.

Figure 5 shows a continuous significant increase in the host serum cytokine profiles of (A) IL-15, (B) IL-18, and (C) M-CSF up to day 25 post-implantation of MiaPaCa-2-eGFP cells, followed by a gradual decrease when the tumor begins to grow at day 30. Interestingly, when OP at the indicated dosages was injected intraperitoneally, all three cytokine profiles were continuously elevated with concomitant expected impeding MiaPaCa-2-eGFP pancreatic tumor time-to-progression growth rate compared to the untreated cohort.

It is interesting to note here that IL-15 has similar functions to IL-2 in that it can stimulate the activation of T-cells, the generation of cytotoxic effector T-cells, and the activation of NK cells [39], all of which can reduce tumor growth. Similarly, IL-18 has been reported to have potent anti-tumor effects. These effects were found to be mediated by effector T cells and NK cells that are in part mediated by apoptosis induction and angiogenesis inhibition [40,41]. Also, M-CSF’s anti-tumor effects were found to be more effective early on in tumor progression. However, they were related to the dosage of M-CSF used [38] and indirectly related to the host immune system.

The data presented in Figure 5 reveal that the cancer cells at the early stage of tumor development orchestrated the angiogenic IL-15, Il-18, and M-CSF cytokines to establish neovasculature to form blood vessels for tumor growth. However, they decreased their effectiveness when the tumor grew at day 35. Interestingly, OP upended this angiogenic staging of tumorigenesis and induced anti-tumor growth. Blood supply and, therefore, vasculature are necessary for tumor growth. This is a complex process, with neoangiogenesis being the only possible mechanism. Metastatic spread via blood vessels is the major mechanism but not the only one (e.g., lymphatic spread).

### 3.4. Dose-Dependent Effect of Oseltamivir Phosphate (OP) on Angiogenic and Proinflammatory Leukemia Inhibitory Factor (LIF), Monokine Induced by IFN-γ (MIG), and Macrophage Inflammatory Protein-2 (MIP-2) Cytokine Profiles Affecting Pancreatic MiaPaCa-2-eGFP Tumor Growth in Heterotopic Xenograft Mice

LIF plays an interesting but complex role in tumor development and progression. Notably, LIF promotes the development and progression of many types of solid tumors. Overexpression of LIF induces the proliferation of cultured human cancer cells and increases the growth of xenograft tumors formed by many human tumor cells [42]. During the early stages of tumor development, chemokines have been shown to reshape the immune landscape of the tumor microenvironment. For example, a CXCL9 monokine induced by interferon-γ (MIG) is produced during inflammatory conditions by myeloid cells within the tumor microenvironment [43]. Overexpression of CXCL9 MIG can also reduce tumor progression and metastasis via the inhibition of angiogenesis. Another CXC chemokine macrophage inflammatory protein-2 (MIP-2) has been shown to promote the outgrowth of colorectal liver metastasis by enhancing angiogenesis, tumor growth, and tumor cell migration [44].

Figure 6 shows a continuous significant increase in the host serum cytokine profiles of (A) LIF, (B) MIG, and (C) MIP-2 up to day 25 post-implantation of MiaPaCa-2-eGFP cells, followed by a marked decrease when the tumor begins to grow after day 30. From the data following OP treatment of mice at day 35, several important questions are raised on LIF cytokine profiles. It is known that LIF expression can be induced under many conditions, but it is unclear why the induction of LIF under OP treatment remains high in the serum. One possibility is that there are many different transcriptional factors in different cell/tissue types during the tumor development stages. Another possibility is that the induction of LIF by different transcription factors may be influenced by different kinetics in inducing LIF, which may lead to a transient increase or sustained increase in LIF, as depicted in Figure 6A. Interestingly, Figure 6A shows the differential regulation of LIF at the protein level in the serum during tumor progression following OP treatment.

As shown in Figure 6B, OP treatment of mice at day 35 maintained high levels of MIG, which has been demonstrated to reduce tumor progression and metastasis via the inhibition of angiogenesis. On the other hand, OP treatment showed a marked diminution of MIP-2 chemokine after day 35, which will upend angiogenesis, tumor growth, and tumor cell migration.

### 3.5. Dose-Dependent Effect of Oseltamivir Phosphate (OP) on Angiogenic Vascular Endothelial Growth Factor (VEGF), Fibroblast Growth Factor β (FGFβ), and Platelet-Derived Growth Factor-BB (PDGF-BB) Cytokine Profiles Affecting Pancreatic MiaPaCa-2-eGFP Tumor Growth in Heterotopic Xenograft Mice

Tumor vasculature requires stringently balanced VEGF signaling to induce sufficient productive angiogenic factors for tumor development. The data in Figure 7A indicate that OP treatment of tumor-bearing mice upended tumor-induced vasculature via an unknown mechanism. Pasula et al. [45] reported that prolonged VEGF levels, as depicted in Figure 7A, and signaling in the absence of endothelial epsins 1 and 2, produce leaky, defective tumor angiogenesis and, thus, contribute to the retardation of tumor growth. The epsin proteins are a family of ubiquitin-binding endocytic clathrin adaptor molecules. Pasula et al. [45] found that both epsins-1 and -2 knockout mice had highly disorganized vascular structures with increased vascular permeability in tumors. It was proposed that the increased non-productive tumor angiogenesis and a retarded tumor growth was due to the increased VEGFR2 signaling.

Figure 7B depicts fibroblast growth factor-β. FGF-β receptor plays a crucial role in several cellular activities, including proliferation, differentiation, and survival. Dysregulation of the FGF-β receptor signaling pathway is implicated in numerous human cancers, making FGFR a prominent therapeutic target. Increased expression of both FGF proteins and their receptors is often associated with the development of chemoresistance, limiting the effectiveness of currently used anti-cancer therapies [46]. The data depicted in Figure 7B show that there is a decrease in the FGF cytokine levels when the tumor starts to develop, but OP treatment increases FGF levels in the serum. It is unclear what the molecular mechanisms are involved in FGF cytokine levels in response to tumor development.

Li et al. [47] reported that PDGF-BB signaling promotes the malignancy of pancreatic cancer via the YAP activation signaling pathway in that the RhoA/PP-1 cascade was involved in the PDGF-BB-induced dephosphorylation of YAP. Targeting PDGFR represses YAP activity and induces tumor apoptosis. The data depicted in Figure 7C suggest that tumor cells require PDGF-BB to promote the development of a tumor, after which PDGF-BB declines at day 35. Notably, OP treatment of mice maintains a low level of PDGF-BB in the serum to initiate and promote tumor apoptosis and regression.

## 4. Discussion

The present study reveals for the first time how cancer cells can orchestrate oncogenetic contemporaneous inflammatory, angiogenic, fibrogenic, and angiostatic cytokine profiles to develop tumor neovascularization, progression, and metastasis. Here, blood samples were collected before MiaPaCa-2-eGFP pancreatic cancer cell implantation and every week until the end point of this study. The extracted serum from the blood of each mouse at different time points during tumor development was analyzed using a Bio-Plex microarray analysis and a Bio-Plex 200 system for proinflammatory (IL-10, IFN-γ, and TNF-α) and angiogenic and fibrogenic (IL-15, IL-18, basic FGF, LIF, M-CSF, MIG, MIP-2, PDGF-BB, and VEGF) cytokines. Here, we find that pancreatic cancer cells orchestrated host angiogenic, fibrogenic, and proinflammatory cytokine profiling as well as the endothelial CD31+ cells in the tumor microenvironment by significantly reducing host angiostatic and proinflammatory cytokines that restrain tumor development and increasing those for tumor growth. Mice treated with oseltamivir phosphate (OP) at different doses slowed the tumor progression with a concomitant direct effect on the cytokine profiling during tumor progression. The data suggest an entirely new regulation by pancreatic cancer cells for tumor development. The findings identify for the first time how pancreatic cancer cells regulate host cytokine profiling and endothelial CD31+ cells to orchestrate the initiation of tumor neovascularization and development (Figure 1, Figure 2 and Figure 3). There are four distinct steps in angiogenesis initially involving (1) degradation of basement membrane by proteases; (2) migration of endothelial cells (ECs) into the interstitial space and sprouting; (3) ECs proliferation at the migrating tip; and, lastly, (4) lumen formation, generation of new basement membrane with the recruitment of pericyte, formation of anastomoses, and, finally, blood flow. The present study used fresh blood samples to measure the CD31+ endothelial cells (ECs) during tumor development. The data, for the first time, showed an interesting pattern of CD31+ EC migration during tumor development, and its decline. These results are consistent with the concept that angiogenesis starts before tumor growth and stops while the tumor explodes in size.

The data depicted in Figure 4, Figure 5, Figure 6 and Figure 7 show the time-to-tumor progression of host serum cytokine profiles during pancreatic MiaPaCa-2-eGFP tumor development in RAG2xCγ double mutant mice. Notable, IL-15, IL-18, IL-1β, and TNFα in the host serum, potentially derived from the tumor microenvironment through various pathways, promoted sustained angiogenesis involving endothelial cells as reported elsewhere [48]. In contrast, the anti-angiostatic MIG, M-CSF, IL-10, and IFNγ cytokines in the host serum revealed a dramatic and significant decrease after day 5 post-implantation of cancer cells. Interestingly, the percent of host plasma CD31+ endothelial cells dramatically declined during tumor progression, indicating the cancer cells’ recruitment ability of endothelial cells to form tumor vascular networks (Figure 4D–F).

Gilmour et al. [29] reported that therapeutic targeting of Neu1 with OP upended pancreatic cancer growth and metastatic spread in heterotopic xenografts of MiaPaCa-2-eGFP tumors growing in RAGxCγ double mutant mice. The report also revealed that OP-treatment of MiaPaCa-2 cells exhibited a reduction of phosphorylation of EGFR-Tyr1173 and downstream phosphorylation of Akt-Thr308, NFκBp65-Ser311, Stat1-Tyr701, and PDGFRα-Tyr754 but, interestingly, an increase in phospho-Smad2-Ser465/467 and -VEGFR2-Tyr1175. The findings of this report identify a novel promising alternate treatment for human pancreatic cancer. Due to its neuraminidase-inhibiting qualities, OP has also been shown to shut down tumor growth and anti-proliferative and anti-metastatic effects on various other cancerous cells [49]. Here, we investigated the inhibition of tumor growth rate following one, four, and five intraperitoneal injections of OP dosages of either 50, 100, or 200 mg/kg at day 42 post-implantation of MiaPaCa-2-eGFP cells. The data revealed that OP with five injections inhibited 40–50% of the tumor growth compared to the untreated cohort (Figure 2A–C).

Taken together, we questioned if OP can also influence the proinflammatory, angiostatic, and fibrogenic cytokine profiling during tumor progression. Notably, the data presented in Figure 5, Figure 6 and Figure 7 suggest that the cancer cells at the early stage of tumor development orchestrated the angiogenic (IL-15, Il-18, M-CSF, VEGF, and FGFβ), proinflammatory (LIF, MIG, and MIP-2), and fibrogenic (PDGF-BB) cytokines to establish neovasculature to form blood vessels for tumor growth but decrease their effectiveness when the tumor starts to grow at day 35. Interestingly, OP impeded these cytokine angiogenic stagings of tumorigenesis at day 35 to induce anti-tumor growth until the end point of the study at day 65. IL-15 is a cytokine known to stimulate cell differentiation proliferation and function as an anti-apoptotic factor in dendritic cells, monocytes, and macrophages [50]. IL-15 activates proliferative mechanisms through various T-cell-associated pathways [50]. In IL-15 levels, little other than a small increase back to its previous peak and a significant decrease in tumor volume by the end of the trial was seen.

IL-18 is a potent inflammatory cytokine that is secreted by macrophages and NK cells. IL-18 can also induce hypoxia-inducible factor-1α (HIF-1α), which mediates tumor progression and can encourage tumor metastasis [32]. IL-18 has been shown to upregulate CD44, VEGF, and MMP-9 in an NF-κB-dependent manner, which increases cancer’s metastatic properties [32]. Feng et al. [51] observed that upregulation of IL-18 inhibited proliferation in colon cancer. In addition, a meta-analysis has demonstrated that downregulation of IL-18 is associated with poor prognosis in carcinomas [52]. IL-18 is highly involved in host immune response as well as autoimmunity [53]. This suggests that IL-18 has the potential to be a valuable tool in immunotherapy using CAR T cells (Chimeric Antigen Receptor T cells) [54].

The proinflammatory cytokine IL-1β is produced mainly by blood monocytes, tissue macrophages, skin dendritic cells, and brain microglia [55]. Once released, IL-1β activates the IL-1R1 receptor, which leads to downstream signaling activation of several factors linked to NF-κB- mediated transcription of proinflammatory cytokines [55]. Tulotta et al. [55] also suggested that in combination with osteoprotegerin (OPG), IL-1β has a role in breast cancer metastasis mediation. IL-1β has been shown to stimulate the production of MMP-9 and the proto-oncogene tyrosine-protein kinase Src, which heavily encourages cancer metastasis and proliferation [55]. It has also been found to play a profound role in upregulating VEGF to cause endothelial cell migration and tube formation, which is very favorable for angiogenic growth [55].

Tumor necrosis factor alpha (TNF-α) is a proinflammatory, cell survival regulating cytokine that is involved in activating signaling pathways such as NF-κB and c-Jun N-terminal kinase (JNK) [35]. These pathways are involved in many anti-apoptotic, proliferative, invasive, metastatic, and tumor angiogenic pathways [35]. Single nucleotide polymorphisms at position −308 of the TNF-α promoter region have been associated with an increased risk of many types of cancers [35]. Furthermore, TNF-857T alleles are associated with an increased risk of B-cell lymphoma [35].

MIG, on the other hand, is an inflammatory angiostatic CXC chemokine [56,57]. Angiostatic CXC chemokines have a variable amino acid in between two cysteine residues and inhibit the growth of blood vessels during wound repair [56,57]. The chemokines’ angiostatic properties give them the ability to prevent tumor growth as well as prevent tumor-associated vessel expansion [57]. Chemokines are also known to stimulate cell migration [56]. MIG, specifically, is known to bind to activated T cells, causing their stimulation, polarization, and locomotion, as well as activating NK cells [58,59,60]. It has been reported that MIG prevents endothelial cell proliferation through changes to growth factor receptor function [60]. Sgadari et al. [58] found that MIG was upregulated in mice with regressed Burkitt’s lymphoma tumors. Rainczuk et al. [61] also reported that angiostatic chemokines prevent the progression of epithelial ovarian cancer.

Interestingly, the inflammatory angiostatic cytokines MIG, M-CSF, IL-10, and IFNγ were found to decrease in time-to-tumor progression (Figure 6). For the first time, pancreatic cancer cells can regulate and reduce these angiostatic cytokines for tumor progression.

Macrophage colony-stimulating factor (M-CSF) is an angiogenic cytokine expressed in many cells and tissues [62,63]. M-CSF is known to produce progenitor cells from bone marrow [63,64]. It has also been reported to play a role in mononuclear phagocyte development and proliferation [63,65]. As well as its involvement in the immune system, M-CSF is involved in ovarian cancer, breast cancer, prostate cancer, and Hodgkin’s lymphoma [63,66,67]. However, according to Yi et al. [68], due to M-CSF’s ability to manage macrophage differentiation, it initiates and enhances macrophage-mediated cytotoxicity against tumor cells and stimulates phagocytosis [68]. As a result of M-CSFs’ ability to do so, cancer cells must find a way to decrease their activity or mutate their actions to their advantage. Since M-CSFs often act through TAMs, perhaps cancerous cells find a way to inhibit the production of TAMs to prevent the activity of M-CSF and allow their growth to progress [69].

IL-1β may play an important role in aiding cancer metastasis, angiogenesis, and growth. IL-1β is often released in response to various PAMPs and DAMPs, inflammasomes activated through various infections and activation of the P2X7 receptor, whose dysregulation has been linked to increased poor outcomes in cancerous growths [70]. Supporting evidence from Bent et al. [71] suggests that increased IL-1β levels are associated with the suppression of adaptive immunity, tumor promotion, and metastasis, supporting our evidence.

IL-10 is an immune-regulatory cytokine that has immunosuppressive and anti-angiogenic functions as it is produced by macrophages, T lymphocytes, and NK cells [72]. IL-10 inhibits tumorigenesis via downregulation of VEGF, IL-1β, TNF-α, IL-6, and MMP-9 and inhibition of NF-κB [72]. As demonstrated in Figure 4F, IL-10 is downregulated far before tumor volume begins to grow, likely to allow for less immunosuppression and more angiogenesis at the site to enable tumor growth. Moreover, an even sharper decline in IL-10 levels is demonstrated as tumor volume begins to grow. This pattern is completely logical as inhibition of IL-10 allows for less immunosuppression, more angiogenesis, and more VEGF, IL-1β, TNF-α, IL-6, MMP-9, and NF-κB activation, which encourages tumor volume. The findings of Tanikawa et al. [73] further backed our data that states that IL-10 suppression promotes tumor growth and development, as demonstrated in our findings.

IFN-γ is a pro-apoptotic, anti-proliferative, anti-tumor interferon that has been shown to inhibit angiogenesis in tumor tissue, stimulate M1 proinflammatory activity to overcome tumor progression and support various functions in the tumor microenvironment [74]. At high doses, IFN-γ takes an anti-tumorigenic role. In contrast, at low doses, it helps the tumor acquire many pro-tumorigenic characteristics like signaling insensitivity, downregulation of major histocompatibility complexes, and various checkpoint inhibitors [74]. In Figure 4H, IFN-γ levels start high and steadily decline as tumor growth begins. On day 27, as tumor volume begins to rise, there is a sharp decrease in IFN-γ levels, signifying that IFN-γ is helping the tumor acquire pro-tumorigenic properties and promoting its growth and development.

FGF-β is a growth factor whose excessive mitogenic signaling may promote cancer progression and increase angiogenic potential, encouraging metastatic cancerous phenotypes and whose presence is crucial to the tumor microenvironment [75]. FGF-β activates receptors that lead to endothelial cell proliferation, migration, and angiogenesis [75]. FGF-β expression in endothelial cells has been shown to control the expression of genes in the cell cycle, differentiation, adhesion, and cell survival, making it a crucial player in promoting endothelial cell angiogenesis [75]. In Figure 7B, we note high levels of FGF-β, which indicates an adhesive, proliferative phenotype. However, when the tumor volume begins to grow aggressively, FGF-β levels drop quickly, which is beneficial for the tumor since FGF-β’s adhesive properties make metastasis difficult. This is of crucial importance because it demonstrates OP’s efficiency in shrinking tumors.

LIF belongs to the interleukin-6 family of cytokines [76]. This cytokine can act as either a proinflammatory or an anti-inflammatory cytokine [77]. Depending on the cell type and other factors, LIF can promote the progression of some tumors while promoting cell death in others [77,78]. Generally, LIF is thought to respond to injured tissue as a result of interactions, which LIF mediates, between the nervous system and the immune system [76,77]. Moreover, LIF is known to prevent the differentiation of stem cells [79]. The data depicted in Figure 4 show that tumor growth results in a decrease in LIF levels. In pancreatic cancer cells, LIF has been reported to activate the JAK/STAT3 pathway, which upregulates genes involved in cell proliferation and survival [80,81]. We observed a downregulation of LIF production in all mice when the tumor began to grow. This is likely due to LIF’s inhibitory effects on differentiation.

Nevertheless, LIF also plays a role in pancreatic cancer cell proliferation. Based on the current research, LIF aids in tumor growth and invasiveness by preventing differentiation and activating the JAK/STAT3 pathways. However, we observed that the tumor signals to the host to downregulate the production of LIF as it grows. These data do not support the current literature regarding LIF and its involvement in cancer progression and, therefore, need to be further elucidated to determine the underlying mechanism.

Similar trends were observed for pro- or anti-inflammatory macrophage colony-stimulating factor (M-CSF), monokine induced by IFN- (MIG), and macrophage inflammatory protein 2 (MIP-2) (Figure 6). In untreated mice, LIF levels rose to a mean level of approximately 30 pg/mL before rapidly decreasing to below 20 pg/mL as tumor volume rapidly increased. In OP-treated mice, at all concentrations, LIF levels peaked at the same concentration; however, they decreased less drastically to approximately 25 pg/mL. Also, LIF levels decreased to these lower concentrations over about 30 days in the OP-treated mice, whereas in the untreated mice, the decrease in concentration occurred over 10 days.

Interestingly, MIP-2 levels had much more variable trends. MIP-2, like MIG, is a proinflammatory chemokine [82]. It has also been reported to promote metastasis as well as angiogenesis [83]. Generally, it is involved in liver injury repair through the activation of neutrophils and the release of inflammatory mediators [84]. The data suggest that tumor signals downregulate the host MIP-2, which prevents attacks from neutrophils and increases the survival of cancer cells.

Furthermore, we investigated the fibrogenic cytokine platelet-derived growth factor-BB (PDGF-BB) (Figure 7). It was observed that there was a general downregulation of PDGF-BB in untreated and OP-treated mice. As a potent mitogen and chemoattractant that induces cellular proliferation, PDGF-BB enhances survival and migration in various tumors, including bladder, breast, and cervical carcinoma [85]. In particular, the PDGF-BB disulfide-bonded dimeric isoform executes cellular signaling through the α- or β-tyrosine kinase receptor (PDGFRα and PDGFRβ, respectively) and can activate downstream signaling complexes, including the extracellular signal-regulated kinase (ERK) and phosphoinositide-3-kinase–protein kinase B/Akt (PI3K-PKB/Akt) pathways [86,87]. Within the tumor microenvironment, PDGF-BB mediates tumor and stromal cell receptor phosphorylation in a paracrine or autocrine fashion [88]. Enabling the indirect influence of tumor growth, metastatic dissemination, and drug resistance, the existing literature suggests that tumor-derived PDGF-BB may support tumor angiogenesis via facilitating stromal fibroblast, perivascular, and endothelial cell recruitment and growth [89,90].

Furthermore, PDGF-BB autocrine signaling may support tumorigenesis and influence malignant phenotypes of cell proliferation, epithelial–mesenchymal transition (EMT), energy metabolism, invasion, metastasis, and colonization [87]. This study found that PDGF-BB is downregulated when the tumor begins to increase in size rapidly. However, it would be expected, based on the previous literature, that the tumor would benefit from elevated levels of PDGF-BB. It is interesting to see this downregulation as it would be expected that PDGF-BB would be upregulated due to its ability to induce proliferation, survival, and metastasis, as well as its involvement in angiogenesis. These data do not support the current literature regarding PDGF-BB’s role in carcinoma cancers and, therefore, need to be further examined to elucidate the underlying mechanisms involved.

Vascular endothelial growth factor, like M-CSF, is an angiogenic proinflammatory cytokine. As an endothelial cell-specific mitogen, vascular endothelial growth factor (VEGF) can trigger physiological and pathological angiogenesis, which influences the host’s immune response to tumors and impacts immune cell function in the tumor microenvironment [91,92]. Autocrine and paracrine VEGF signaling within the tumor microenvironment supports tumorigenesis, particularly contributing to the cancer stem cell function and the downstream signaling facilitated by VEGF receptor tyrosine kinases (RTKs) and neuropilins (NRPs) [93]. Essential to endothelial cell remodeling during tumorigenesis, immune-cold tumors exhibit decreased endothelial adhesion proteins but increased VEGF and hypoxia pathways [94].

Cell adhesion molecule-1 (CD31) is a well-known marker of adhesion and accumulation of platelets in endothelial cells [95]. CD31 has been shown to play critical roles in cell proliferation, apoptosis, migration, and invasion [95]. Zhang et al. [95] demonstrated CD31’s ability to promote metastasis by inducing epithelial–mesenchymal transition cells in carcinomas.

In this present study, Figure 1 highlights the potential efficacy of oseltamivir phosphate in increasing the probability of survival for individuals with cancer through a theorized anti-tumorigenesis mechanism. While all concentrations of oseltamivir phosphate had greater probabilities of survival compared to untreated mice, Figure 1 alludes to 100 mg/kg of oseltamivir phosphate having the highest probability of survival for a larger number of days post MiaPaCa-eGFP implantation. This suggests that the concentration of oseltamivir phosphate is an important variable to consider in future studies evaluating oseltamivir phosphate anti-tumorigenesis efficacy.

Similarly, Huang et al. [96] found that oseltamivir significantly inhibited Huh-7 and HepG2 cellular growth and migration, in addition to triggering apoptosis and autophagy in these liver cancer cell lines [49]. While these data contribute to a greater understanding of how oseltamivir phosphate may be used as a potential cancer therapeutic, clinical trials with human subjects are required to ensure its safety and efficacy in a clinical setting.

### 4.1. Clinical Relevance

Understanding the functions of these cytokines and how they are impacted by tumor behavior suggests a potential new tool for therapy targets. Cytokine profiling throughout a patient’s treatment rather than just at the beginning could also provide insight into the progression of the disease and inform decisions on the treatment used. Furthermore, the emphasis placed on OP’s ability to regulate tumor microenvironments by affecting certain cytokines and not others helps us better understand its effectiveness as a future therapeutic agent.

### 4.2. Future Directions

These studies were conducted for a maximum of 70 days, and results may change beyond that time point. However, these findings suggest that incorporating OP into treatment protocols of patients with cancer will reduce tumor volume, metastasis, and proliferation and change the microenvironment to be more manageable. This can improve patient outcomes by reducing disease severity and improving outcomes. This may be an excellent alternative therapy for patients who are not finding success with other treatments.

## 5. Conclusions

Our study demonstrates the complex interaction between tumors and host immune responses, highlighting the tumors’ ability to manipulate cytokine production to create a microenvironment conducive to their survival and growth. Tumors regulate vital cytokines such as IL-18, LIF, MIG, MIP-2, and M-CSF to evade immune detection, promote angiogenesis, and enhance proliferation. In addition, PDGF-BB was downregulated in untreated and OP-treated mice, contrary to expectations based on its known role in other cancers, highlighting the need to examine its role in pancreatic cancer further.

This study clearly shows that OP has various effects on different cytokines. Specifically, OP treatment helps to maintain higher levels of cytokines and regulate harmful fluctuations, enhancing anti-tumor immunity and stopping tumors’ uncontrolled growth.

Understanding the cytokine dynamics can provide valuable insights into new treatment strategies. Manipulating cytokine production against tumors could improve the effectiveness of immunotherapy and other cancer treatments. These findings lay the foundation for future studies to elucidate further the mechanisms of cytokine regulation in the tumor microenvironment and develop interventions that can better control tumor progression and improve prognosis in patients with pancreatic cancer.

## Figures and Tables

**Figure 1 cells-13-01739-f001:**
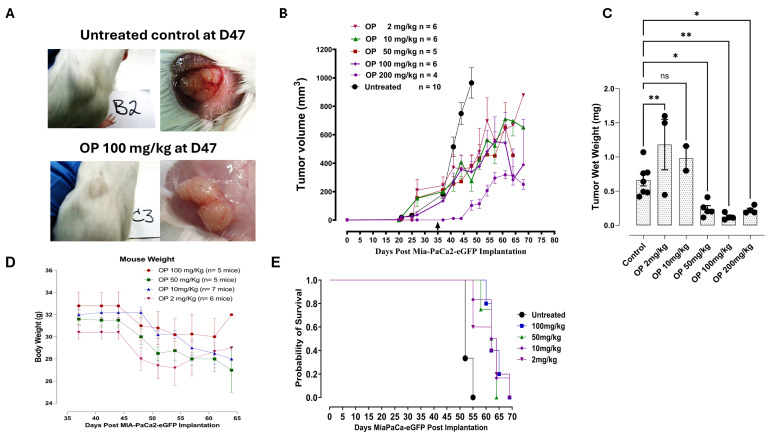
(**A**) Necropsy tumor images at day 47 from RAG2xCγ double mutant mice implanted with MiaPaCa-2-eGFP pancreatic cancer cells following intraperitoneal daily treatment with soluble 100 mg/kg OP. (**B**) Time-to-tumor progression of daily OP treatment started at day 35 (arrow) at indicated dosages, (**C**) tumor wet weight, (**D**) body weight of cohorts following OP treatment at indicated dosages. The animals were monitored daily for health by a certified veterinarian technician. (**E**) The probability of survival for days post-implantation of MiaPaCa-2-eGFP of untreated mice and mice treated with OP concentrations at 2 mg/kg, 10 mg/kg, 50 mg/kg, and 100 mg/kg. The log-rank (Mantel–Cox) test was used to test the probability that the survival curves were significantly different using the Chi-square. The probability of survival of the OP-treated mice compared to the untreated was significant, *p* < 0.0050. The one-way ANOVA Fisher test comparisons with 95% confidence use asterisks to indicate statistical significance. * *p* < 0.05, ** *p* < 0.001.

**Figure 2 cells-13-01739-f002:**
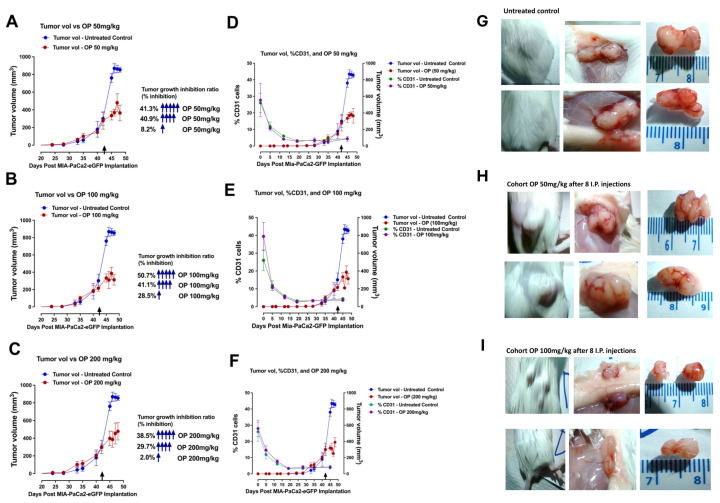
(**A**–**C**) Tumor growth inhibition rate following one, four, and five intraperitoneal injections of oseltamivir phosphate (OP) dosages of 50, 100, and 200 mg/kg at day 42 post-implantation of MiaPaCa-2-eGFP cells. The arrows indicate the number of intraperitoneal injections of OP and the percent inhibition of tumor growth. (**D**–**F**) Flow cytometry results of the presence of the characteristic mouse CD31+ endothelial cells in the blood during MiaPaCa-2-eGFP tumor-bearing RAG2xCγ double mutant mice which had received various treatments of oseltamivir phosphate (OP) of 50, 100, and 200 mg/kg at day 42 (arrow). Blood was collected retro-orbitally at indicated time points, and the percentage of CD31+ endothelial cells was measured using flow cytometry. Results were graphically depicted as line graphs with points depicting the mean± SEM, n = 4 mice for tumor volume and the percentage of CD31+ cells of each group. (**G**–**I**) Necropsy tumors after eight IP injections of indicated dosages of OP expressed subcutaneously, exposed under the skin, showing the extent of neovasculature and tumor size.

**Figure 3 cells-13-01739-f003:**
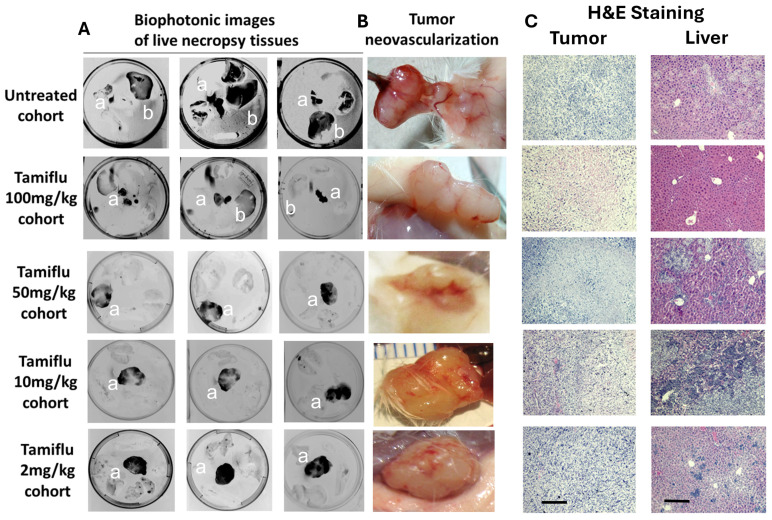
(**A**) Biophotonic images of live, non-processed necropsy of a—tumor, b—liver, from the untreated cohort, 2 mg/kg OP cohort, 10 mg/kg OP cohort, 50 mg/kg OP cohort, and 100 mg/kg OP cohort. The dark tissue images represent the invasion of MiaPaCa-2-eGFP cells. (**B**) Representative images of tumor neovascularization from untreated cohort and OP-treated cohorts. (**C**) H&E staining of tumor and liver from untreated cohort and OP-treated cohorts. The scalebar represents 100 µm.

**Figure 4 cells-13-01739-f004:**
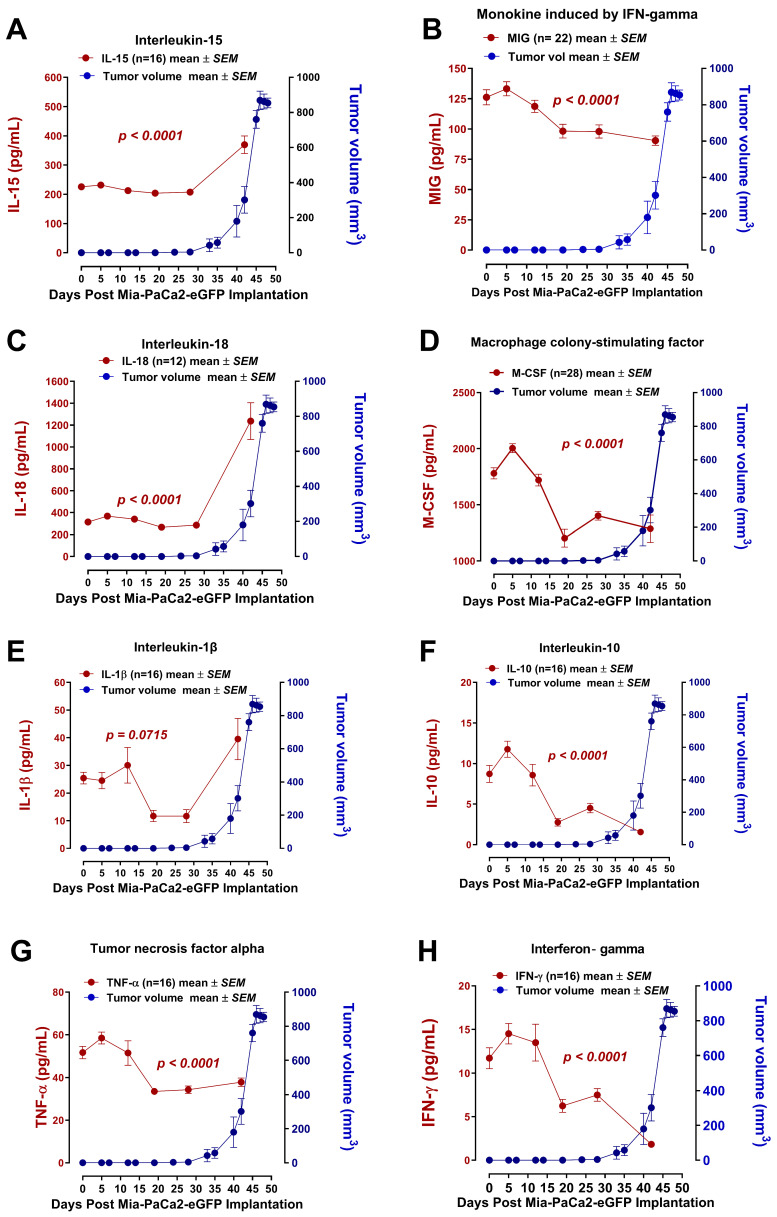
Time-to-tumor progression of host serum cytokine profiles during pancreatic MiaPaCa-2-eGFP tumor development in RAG2xCγ double mutant mice. (**A**) Interleukin 15 (IL-15), (**B**) monokine induced by interferon-gamma (MIG), (**C**) interleukin-18, (**D**) macrophage colony-stimulating factor (M-CSF), (**E**) interleukin- 1β (IL-1β), (**F**) interleukin-10 (IL-10), (**G**) tumor necrosis factor-alpha (TNF-α), and (**H**) interferon-gamma (IFN-γ) cytokine levels were measured (pg/mL) in host serum before and every five (5) days after MiaPaCa-2-eGFP implanted xenografts in RAG2xCγ double mutant mice for 45 days. After MiaPaCa-2-eGFP implantation, tumors grew for 50 days, and tumor volumes were measured in mm^3^. Blood was collected retro-orbitally, and cytokine levels were measured using a magnetic bead-based Luminex Bio-Plex microarray mouse cytokine assay. Results are depicted as mean ± standard error of the mean ± SEMwith indicated mouse numbers. A one-way ANOVA was used to test for linear trends and measure statistical significance.

**Figure 5 cells-13-01739-f005:**
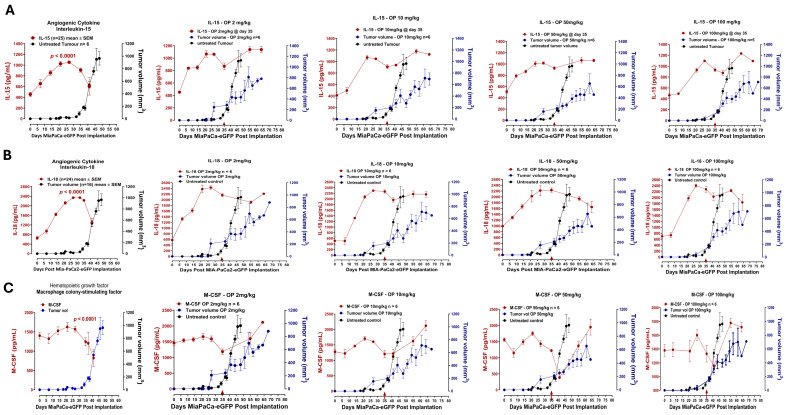
Dose-dependent effect of oseltamivir phosphate (OP) on (**A**) interleukin-15 (IL-15), (**B**) IL-18, and (**C**) macrophage colony-stimulating factor (M-CSF) angiogenic cytokine profiles affecting pancreatic MiaPaCa-2-eGFP tumor growth in heterotopic xenograft mice. Host serum cytokine profiles (pg/mL) were measured in a time-to-tumor progression of pancreatic MiaPaCa-2-eGFP tumor development. Cytokine levels in the serum were analyzed using a magnetic bead-based Luminex Bio-Plex microarray mouse cytokine assay. Results are depicted as mean ± standard error of the mean (SEM) with indicated mouse numbers. Tumor volumes (mm^3^) of untreated and OP-treated mice were plotted concomitantly with cytokine profiles. OP treatment at indicated different dosages was injected after day 35 post-implantation (arrow). One-way ANOVA to test for linear trend was used to measure statistical significance, *p* < 0.0001 at indicated n values of untreated mice for groups tested.

**Figure 6 cells-13-01739-f006:**
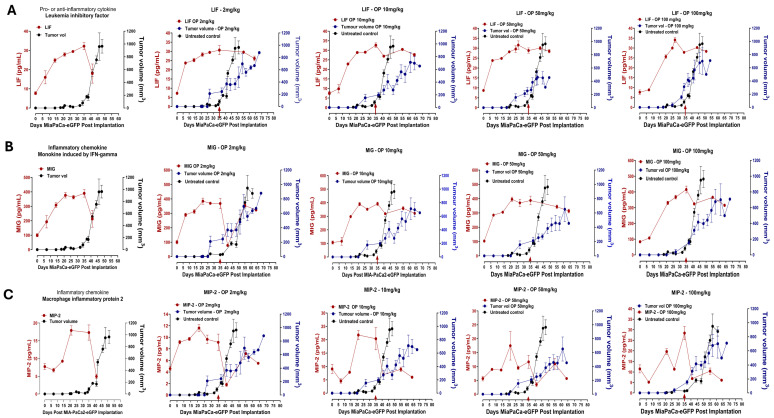
Dose-dependent effect of oseltamivir phosphate (OP) on (**A**) leukemia inhibitory factor (LIF), (**B**) monokine induced by IFNγ (MIG), and (**C**) macrophage inflammatory protein-2 (MIP-2) angiogenic and proinflammatory cytokine profiles affecting pancreatic MiaPaCa-2-eGFP tumor growth in heterotopic xenograft mice. Host serum cytokine profiles (pg/mL) were measured in a time-to-tumor progression of pancreatic MiaPaCa-2-eGFP tumor development. Cytokine levels in the serum were analyzed using a magnetic bead-based Luminex Bio-Plex microarray mouse cytokine assay. Results are depicted as mean ± standard error of the mean (SEM) with indicated mouse numbers. Tumor volumes (mm^3^) of untreated and OP-treated mice were plotted in concomitance with cytokine profiles. OP treatment at indicated different dosages was injected after day 35 post-implantation (arrow). One-way ANOVA to test for linear trend was used to measure statistical significance, *p* < 0.0001 at indicated n values of untreated mice for groups tested.

**Figure 7 cells-13-01739-f007:**
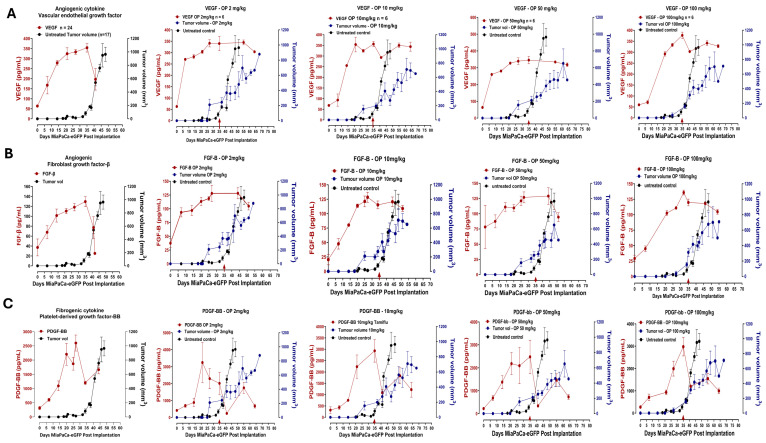
Dose-dependent effect of oseltamivir phosphate (OP) on (**A**) vascular endothelial growth factor (VEGF), (**B**) fibroblast growth factor β (FGFβ), and (**C**) platelet-derived growth factor-BB (PDGF-BB) angiogenic cytokine profiles affecting pancreatic MiaPaCa-2-eGFP tumor growth in heterotopic xenograft mice. Host serum cytokine profiles (pg/mL) were measured in a time-to-tumor progression of pancreatic MiaPaCa-2-eGFP tumor development. Cytokine levels in the serum were analyzed using a magnetic bead-based Luminex Bio-Plex microarray mouse cytokine assay. Results are depicted as mean ± standard error of the mean (SEM) with indicated mouse numbers. Tumor volumes (mm^3^) of untreated and OP-treated mice were plotted in concomitance with cytokine profiles. OP treatment at indicated different dosages was injected after day 35 post-implantation (arrow). One-way ANOVA to test for linear trend was used to measure statistical significance, *p* < 0.0001 at indicated n values of untreated mice for groups tested.

## Data Availability

All data needed to evaluate this paper’s conclusions are present. The preclinical datasets generated and analyzed during the current study are not publicly available but are available from the corresponding author upon reasonable request. The data will be provided following the review and approval of a research proposal, statistical analysis plan, and execution of a data sharing agreement. The data will be accessible for twelve months for approved requests, considering possible extensions; contact szewczuk@queensu.ca for more information on the process or to submit a request.

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
