# Peer review of "Contemporaneous Inflammatory, Angiogenic, Fibrogenic, and Angiostatic Cytokine Profiles of the Time-to-Tumor Development by Cancer Cells to Orchestrate Tumor Neovascularization, Progression, and Metastasis"

_cells, 2024, doi:10.3390/cells13201739_

Round 1

Reviewer 1 Report (Previous Reviewer 2)

Comments and Suggestions for Authors

The resubmitted form is better and completed according to my comments.

Author Response

Comments and Suggestions for Authors

The resubmitted form is better and completed according to my comments.

Author response: Thank you for the comment and support of the study.

Submission Date

21 September 2024

Date of this review

08 Oct 2024 11:47:19

Reviewer 2 Report (New Reviewer)

Comments and Suggestions for Authors

Thanks for inviting me to review this article, which was interesting. Seen from the different colors of words, I inferred that this was a resubmitted paper. In particular, the reviewer appreciated the writing of Discussion. This study was informative, which could be considered for publication in Cells, after a further Minor Revision. Detailed comments:

1.      Please add some critical data in the Abstract. For example, how did the expression of pro-inflammatory, angiogenic and fibrogenic biomarkers change?

2.      Please state the p<0.05 as the threshold of significant difference in Section 2.11.

3.      The name of X-axis was covered in Figure 1B.

4.      According to Figure 1D, the weight of mice was continuously reducing. Please comment on the safety issue.

5.      The scale bars should be added in Figure 3C.

Author Response

Comments and Suggestions for Authors

Thanks for inviting me to review this article, which was interesting. Seen from the different colors of words, I inferred that this was a resubmitted paper. In particular, the reviewer appreciated the writing of Discussion. This study was informative, which could be considered for publication in Cells, after a further Minor Revision. Detailed comments:

  1. Please add some critical data in the Abstract. For example, how did the expression of pro-inflammatory, angiogenic and fibrogenic biomarkers change?
  2. Please state the p<0.05 as the threshold of significant difference in Section 2.11.
  3. The name of X-axis was covered in Figure 1B.
  4. According to Figure 1D, the weight of mice was continuously reducing. Please comment on the safety issue.
  5. The scale bars should be added in Figure 3C.

Author response: Thank you for the comment and support of the study. We have made the correction as suggested in the revised manuscript.

Submission Date

21 September 2024

Date of this review

14 Oct 2024 10:21:54

This manuscript is a resubmission of an earlier submission. The following is a list of the peer review reports and author responses from that submission.

Round 1

Reviewer 1 Report

Comments and Suggestions for Authors

REVIEWER #1

Comments and Suggestions for Authors

The manuscript claims to provide novel mechanisms for pancreatic tumor progression, especially tumorangiogenesis. While the approach of analysis cytokine profiles is interesting, the manuscript has severe flaws.

This manuscripts misses the fact, that systemic analysis (like blood) cannot reflect the TME in full. It solely analyses blood sample. Especially angiogenesis is a local reaction (within the tumor).

Author response: Thank you for the comment. We have revised the title of the manuscript to reflect the data of the study, namely, “Contemporaneous Inflammatory, Angiogenic, Fibrogenic and Angiostatic Cytokine Profiles of the Time-to-Tumor Development and Progression Orchestrated by Cancer Cells to Develop Tumor Neovascularization, Progression and Metastasis.” 

Reviewers response: Thank you for the changes

 It is correct that blood analysis does not reflect the TME fully. In the revised manuscript, we have included new data demonstrating the necropsy tumor bloody neovascularization of the untreated control cohort and the OP-treated cohorts in the new Figures 1-3. In Figure 3, biophotonic images of live necropsy of a—tumor, b—liver, c—lung, d—spleen and e—heart from the untreated cohort, 2mg/kg O.P. cohort, 10mg/kg O.P. cohort, 50mg/kg O.P. cohort, and 100 mg/kg O.P. cohort. We have included in Fig 3(B) representative images of tumor neovascularization from untreated cohort and OP-treated cohorts as well as in Fig 3(C) the H&E staining of tumor and liver from untreated cohort and OP-treated cohorts.

The data in Figure 3A reveal the biophotonic images of MiaPaCa-2-eGFP tumors and their migration to the liver, lung, spleen and heart, revealing minimal or negligible metastatic MiaPaCa-2-eGFP of OP-treated cohorts compared to the untreated control. Notably, the tumors from the OP-treated cohorts show ufewer bloody vessels/u compared to the untreated cohort, confirming the O.P. affecting the tumor neovascularization process.  

Reviewers response:  Blood analyses can only show a part of tumor induced changes. Commonly for TME tumor tissue and for immunological changes spleen and lymphnodes are analysed. Results often show differential profiles depending on tissue analysed. Nevertheless changes in blood parameters could be a very important – as surrogate marker for tumor progression and treatment success. Therefore the now included tumor data are essential for this manuscript.   Macroscopic tumor photographs are a starting point for tumor angiogenesis – but the common marker is CD31 immunohistochemistry. Metastasis: Only one tumor cell line has been used in one (highly) immunodeficient mouse strain. Only blood analysis (cytokines and one cell type) have been performed. Given that narrow spectrum, the authors should use more caution in general claims

Author response: Thank you for the comment. The present study was an uextensive analyses /uof the host cytokine profiles based on previously peer-reviewed published data (Gilmour et al. Cellular Signalling 25 (2013) 2587–2603). That article provided the proof-of-evidence for therapeutic targeting of Neu1 with O.P. in impeding human pancreatic cancer growth and metastatic spread in heterotopic xenografts of ueGFP-MiaPaCa-2 tumors/u growing in RAGxCγ double mutant mice.   

Reviewers response: There seems to be a misunderstanding of this reviewer’s comments. The validity of the s. c. MiaPaCa-2 Xenograft model has not been questioned, nor strategy of using O.P.  The comments only refered to making very general claims on (pancreatic) cancer and more or less neglecting the systemic limitations of xenografting. The present study analyzed the host angiogenic, fibrogenic and pro-inflammatory cytokine profiling in the tumor-bearing mice by significantly reducing host angiostatic and pro-inflammatory cytokines that restrain tumor development and increasing those for tumor growth. The data suggest an entirely new regulation by cancer cells for tumor development. The findings identify for the first time how pancreatic cancer cells use host cytokine profiling to orchestrate the initiation and tumor development. Necropsy tumor images reveal the extent of the bloody vasculature of the tumor in new Figures 1-3.  

 Reviewers response: To this reviewer it is still not clear, why blood analysis would detect ‘host cytokines’. It could very well originate from tumor cells.   Here are mso-fareast-font-family:  MsoListParagraphCxSpFirst align

1.      MiaPaca-2 cells express at IL-18 (but not IL-1beta, IL-10) in vitro (Bellone et al. Cancer Immunol Immunother. 2006 Jun;55(6):684-98. doi: 10.1007/s00262-005-0047-0).

2.      VEGF (e. g. Gerber et al. Proc Natl Acad Sci U S A. 2007 Feb 27;104(9):3478-83. doi: 10.1073/pnas.0611492104; Liang WC, Wu X, Peale FV, Lee CV, Meng YG, Gutierrez J, Fu L, Malik AK, Gerber HP, Ferrara N, Fuh G.J Biol Chem. 2006 Jan 13;281(2):951-61. doi: 10.1074/jbc.M508199200) has been shown to be expressed by tumor cells.

3.      Il-2 and IFNgamma are not detectable upon antigen-stimulation in BALB/c rag2 cgc dko mice (generated by Zhao et al. Front Genet. 2019 Apr 30:10:401. doi: 10.3389/fgene.2019.00401)

Therefore the conclusions of changes in host-derived cytokine profliles are still not valid. This does explicitly not negate systemic cytokine changes caused by tumor and treatment (regardless of which cells are producing them).

For cytokine analyses, ‚murine‘ panels have been used. Therefore host has been claimed as source of these cytokines. It should be stated, if crossreactivity against human cytokines has been tested.

Author response: Thank you for the comment. We have stated this crossreactivity not done  in M&M 2.7. Luminex Bio-plex microarray mouse cytokine assay

Reviewers response: Thank you for this addition. Crossreactivity as an issue has been stated independently e. g. in Gerber et al. (Int J Mol Sci. 2018 Dec 1;19(12):3836. doi: 10.3390/ijms19123836.). This states a systematic problem in distinguishing mouse and human cytokines and therefore even in a human to mouse xenograft, conclusions about host vs tumor origin must be very carefully made.Unfortunatley no details on the Bio-plex microarray (e. g. manufacturer, ordering number) have been given by the authors. But in the past Luminex panels may have been reactive to both species – at least there are informations that VEGF and PDGF have.

(Immuno-)Histological analyses or any other approaches for the tumor are completely missing.

Author response: Thank you for the comment.  In the revised manuscript, we have included new data demonstrating the necropsy tumor bloody neovascularization of the untreated control cohort and the OP-treated cohorts in the new Figures 1-3. In Figure 3, biophotonic images of live necropsy of a—tumor, b—liver, c—lung, d—spleen and e—heart from the untreated cohort, 2mg/kg O.P. cohort, 10mg/kg O.P. cohort, 50mg/kg O.P. cohort, and 100 mg/kg O.P. cohort. We have included in Fig 3(B) representative images of tumor neovascularization from untreated cohort and OP-treated cohorts as well as uin Fig 3(C) the H&E staining of tumor and liver from untreated cohort and OP-treated cohorts./u The data in Figure 3A reveal the biophotonic Images of MiaPaCa-2-eGFP tumors and their migration to the liver, lung, spleen and heart, revealing minimal or negligible metastatic MiaPaCa-2-eGFP of OP-treated cohorts compared to the untreated control. Notably, the tumors from the OP-treated cohorts show fewer bloody vessels compared to the untreated cohort, confirming the O.P. affecting the tumor neovascularization process.  

Reviewers response: Thank you for adding H/E stainings and photographs of excised tumors. Macroscopic images of tumors present an impression of tumor vasculature – but the common marker for tumor angiogenesis is immunohistology with CD31 and for vessel maturation SMA. If quantification of tumor vasculature is not available, the statements on tumorangiogenesis should be stated more specific e. g. as ‘typical examples of blood vessels leading to the tumor are shown for each treatment group’. The extent of metastasis is not comprehensible by the photographs shown (see comment below).        5e99; mso-themecolor: text2; mso-themetint: 191; mso-fareast-language: EN-CA"The methods section is missing crucial nformationn (e. g. ‚official‘ mouse strain nomenclature, ‚standard‘ mouse housing in formation (dark: light cycle, food and water administration)).

Author response: Thank you for the comment. >  Heterotopic xenograft mouse model of human pancreatic cancer - The xenograft mice used in the study were immunodeficient with double mutations in the recombinase combining activating gene-2 (RAG2) and the common cytokine recep-tor γ chain (Cγ). These RAG2xCγ mice completely lack lymphoid cells (T-cell, B-cell, and NK-cell deficient), show no spontaneous tumor formation, and have normal hematopoi-etic parameters. uThe mice were made by inter-crossing and were kept in SPF isolators in the Animal Care Facility, Queen’s University, Kingston, Ontario K7L 3N6, Canada./u A colony of mice was established in the facility. uMice were cared for under sterile conditions in micro-isolators or air-filtered cages and were given autoclaved food and water./u The mice used were between 6 and 8 weeks of age. Mice used in the study were approved by the Animal Care Committee at Queen’s University.  

Reviewers response: Thank you for the additional information on the mouse strain. Please add the background strain. Even though T-cells are missing, it could be important (e. g. for reproducablitiy) if it is C57B/6 or BALB/c based. Please indicate the gender of the mice used./ Blood samples have been taken weekly for cytokine analyses. When and how much has been taken for cell analysis?

Author response: Thank you for the comment. >  Mouse blood serum collection. Blood was collected retro-orbitally by penetrating the retro-orbital plexus located behind the eye of the mouse. u200 µL of blood was collected with a capillary tube./u Blood was left to coagulate and was then centrifuged. Collected serum was then freshly frozen at  -80oC and analyzed later for cytokine profiling using Luminex Bio-plex microarray mouse cytokine assay.  

Reviewers response: Thank you for the detailed information. For subcutaneaous tumor cell injection, a cell number of 1x10sup6/sup is given with a cell concentration of 0,5-1 x 10sup6/sup cell/ml. That makes 1-2ml per injection – which is an abnormally high injection volume. Resuspending (incl. washing) cells in saline may reduce cell viability  

Author response: Thank you for the comment. This issue is corrected in the revised manuscript. Puromycin-resistance eGFP-transduced MiaPaCa-2 cell clones were grown in a 75 cm² culture flask at 80-90% confluence. The cells were put into solution using TrypLE Ex-press (Gibco) and washed with sterile saline. The cells in solution were centrifuged for 5 min at 900 rpm. The cell pellet was resuspended in sterile normal saline at 6x10sup6/sup cells/mL for 0.2 mL injection of 1.2x10sup6/sup cells subcutaneously implantation into the right back flank of the mouse. Tumor measurements were taken twice a week.

 Reviewers response: Thank you for the detailed information  mso-fareast-font-family:Another remarkable finding is, that tumor growth can only be detected after about 5 weeks. Endpoints for the experiments must be described – expecially in the ‚survival‘ experiment (humane endpoints?!) Author response: Thank you for the comment. We have added the following in M&M: Humane endpoints of the experimental mice were monitored daily by a registered veterinary practitioner. Mice were euthanized when the tumor volume reached 1000 mm3 or a body weight loss of 15%, according to Animal Care Protocol. Tumor measurements were taken twice a week.

 Reviewers response: Thank you for the information on monitoring and endpoints  mso-fareast-font-family:  The formula given for s. c. tumor volume is unusually written. The commonly used depiction for elipsoid tumors is a name

Author response: Thank you for the comment.   Tumor volumes were analyzed using the following formula: Volume = (width square) x length x 1/2.  

Reviewers response: Thank you Selection of eGFP clones of MiaPaCa-2 is mentioned, but not described. In addition it is unclear, if stable clone(s) were established. It is not clear, why eGFP transfected cells and not the parent cell line were used.

Author response: Thank you for the comment. The manuscript clearly described the selection of eGFP clones of MaPaCa-2 cells. We used the eGPF clone MiaPaCa-2 cells to visualize the tumor progression and metastasis using ubiophotonic imaging/u (Cancer Biology & Genetics Division of the Queen’s Cancer Research Institute).  

 Reviewers response: With the inclusion of biophotonic imaging, the need of using of eGFP cells is clarified.  2.2. eGFP Lentifect purified lentivirus particles  GeneCopoeia Lentifect™ Lentiviral Particles (Cat# LP-EGFP. LV105-0205) are ready-to-use particles. They are produced from a standardized protocol using purified plasmid DNA and the proprietary reagents EndoFectin™ Lenti (for transfection) and Ti-terBoost™ solution. The protocol uses a third-generation self-inactivating packaging system meeting BioSafety Level 2 requirements with the University Biohazard Committee approval. The Lentifect particles include a cytomegalovirus (CMV) promoter for expression of non-tagged eGFP in target cells, and a puromycin resistance marker is used for the sta-bly selection of transduced cells. The ready-to-use lentiviral particles were used for the transduction of MiaPaCa-2 cells.  Briefly, cells were cultured in 6 well tissue culture plates in DMEM medium containing 10% FCS and 5 μg/mL plasmocin. After 24 hrs, after the medium was discarded, 2 mL of 5 μg/mL of media containing polybrene was added to the cells, followed by eGFP lentiviral particles at MOI=6. The plate was mixed and centrifuged at 2500 rpm for 90 min and incubated at 370C in a 5% CO2 humidified incubator for 24 hrs. After washing, the cells were re-cultured in media for an additional 2 days. On day 5, the media containing an optimal 2μg/mL of puromycin, as pre-determined in a cell viability assay, was replaced in the cells. The selection media was added every 40 hrs to expand puromycin-resistance eGFP-transduced MiaPaCa-2 cell clones. uThe transfection efficiency of 90% was determined using fluorescence microscopy (Zeiss Imager M2) and biophotonic imaging (Cancer Biology & Genetics Division of the Queen’s Cancer Research Institute) before implantation into the mice./u  

Reviewers response: Thank you for the detailed description. Based on this information it seems to this reviewer, that a pool of transduced cells – not a selected clone (based on picking single founder cells) were used. mso-fareast-font-family:  Administration of Oseltamirvir is not described sufficiently (carrier/solvent, concentration/volume, administration route, timing of multiple injections).

Author response: Thank you for the comment. We have added the following to clarify this comment: Inhibitors: a name="_Hlk174824162"/aOseltamivir phosphate (O.P.) (La Roche Ltd., Mississauga, Ontario), the active ingredient in Tamiflu, was used at indicated concentrations. To test the efficacy of O.P. at different dosages, mice were I.P. injected with 0.2 mL of O.P. at 2, 10, 50, 100 and 200 mg/kg in sterile phosphate buffered saline (pH 7.4) daily or otherwise indicated starting at day 35 post-implantation.  

Reviewers response: Thank you. / Claims on tumor immune evasion in a pretty much completly immunodeficient mouse seems far fetched, especially given the fact that (murine) tumors are beeing comparatively analysed in differentially immunodeficient/competent mouse strain sets   a name

Author response: Thank you for the comment.   We disagree with this comment. We have included data from biophotonic imaging (Figure 3A). >  The data in Figure 3A reveal the biophotonic images of MiaPaCa-2-eGFP tumors and their migration to the liver, lung, spleen and heart, revealing minimal or negligible metastatic MiaPaCa-2-eGFP of OP-treated cohorts compared to the untreated control. The OP-treated cohorts show minimal tumor neovascularization compared to the untreated cohort (Figure 3B), supporting the biophotonic tumor data in Figure 3A. The present study was an extensive analyses of the host cytokine profiles based on previously peer-reviewed published data (Gilmour et al. Cellular Signalling 25 (2013) 2587–2603). That article provided the proof-of-evidence for therapeutic targeting of Neu1 with O.P. in impeding human pancreatic cancer growth and metastatic spread in heterotopic xenografts of eGFP-MiaPaCa-2 tumors growing in RAGxCγ double mutant mice.    

Reviewers response: This seems to be a misunderstanding. The reviewer’s comment is NOT on the validity of  the model/experiment/treatment including effects on tumor and metastasis, but on claims on ‘immune evasion’ in this immune-deficient setting./ The questions on the methods have to be answered before evaluating the results and discussion sections.

Author response: Thank you for the comment. We have revised the manuscript accordingly mso-fareast-font-family:  Comments on the Quality of English Language. Only minor spelling checks are necessary

Author response: Thank you for the comment. We use Grammarly software to check the quality of writing and plagiarism. The manuscript reveals 6-7% plagiarism, mainly based on the comment wording to describe Materials and Methods.    

Reviewer’s comments on the revised manuscript:    The presented resubmitted manuscript presents interesting data on systemic detection of tumor induces cytokines. These data on blood analysis are of interest for the research community, since commonly tumor tissue and lymphatic tissue is analysed at endpoint or with sophisticated (imaging) methods. Blood analysis allows - even in mice – for a time-course. Therefore finding (or disproving) changes in blood markers for tumor progression and load even in a specific model is of value for the preclincal research community. While for most cytokine and tumor-host interactions, tumor and lymphatic tissue (spleen, lymphnodes) are anlysed and therefore only endpoint data are available, the blood analysis presented in this manuscript have the major advantage of allowing a time-course This reviewer wants to explicitly state, that the comments are not questionning the model, but the extent to which conclusions are drawn. The focus of the reader should be more directed tothe data of blood levels in comparison to tumors size, timing and treatment   This reviewer appologizes, that some of the comments may be repetitive to the reply of this reviewer to the authors comments of round 1   The authors must take the inherent limitations of the heterotopic xenograft model better into account.    The human MiaPaca-2 cell line has to be implanted into immunodeficient mice. Here a mouse, which lacks the most important tumor fighting immune cells, namely T-, B- and NK cells has been used. This is a well established model – yet no this excludes conclusions towards immune evasion.    Unfortunately the conclusion, that the ‚mouse panels‘ used present host response has to be taken with care, if not to be questioned – without further experiments beyond the scope of this manuscript. The reasons are:  MiaPaca-2 cells express at IL-18 (but not IL-1beta, IL-10) in vitro (Bellone et al. Cancer Immunol Immunother. 2006 Jun;55(6):684-98. doi: 10.1007/s00262-005-0047-0).

3.       VEGF (e. g. Gerber et al. Proc Natl Acad Sci U S A. 2007 Feb 27;104(9):3478-83. doi: 10.1073/pnas.0611492104; Liang WC, Wu X, Peale FV, Lee CV, Meng YG, Gutierrez J, Fu L, Malik AK, Gerber HP, Ferrara N, Fuh G.J Biol Chem. 2006 Jan 13;281(2):951-61. doi: 10.1074/jbc.M508199200) has been shown to be expressed by tumor cells.

4.       Il-2 and IFNgamma are not detectable upon antigen-stimulation in BALB/c rag2 cgc dko mice (generated by Zhao et al. Front Genet. 2019 Apr 30:10:401. doi: 10.3389/fgene.2019.00401)

Information on tumor data has improved. Presentation of tumor size and H/E stainings are important information, but Figures must be improved (see details below). Macroscopic images of tumors present an impression of tumor vasculature – but the common marker for tumor angiogenesis is immunohistology with CD31.

Line 30: ‘mechanisms unknown’: The statement suggests that there is no knowledge on tumorangiogenesis, TME etc. It would be helpful to rephrase that additional information/analysis is still needed. (same for line 38).

Line 30: ‘TME surrounds tumors’, the term TME generally also includes tumor stroma and immune cells – within the tumor node. It is true, though, that tumor cells are embedded in TME.

Lines 46-52: It is unclear, how this data prove that host produced cytokines are influenced by MiaPaCa-2 cells. It is true though, that the systemic cytokine profile is changed (= within the host). Injection of tumor cells is a ‘colonisation’ not a tumor ‘initiation’ model. Cytokines including some of the ones included in the panels used in this study have been analysed in pancreatic cancer before (e. g. two references directly relevant to these experiments: Bellone et al. Cancer Immunol Immunother. 2006 Jun;55(6):684-98. doi: 10.1007/s00262-005-0047-0); Gerber et al. Proc Natl Acad Sci U S A. 2007 Feb 27;104(9):3478-83. doi: 10.1073/pnas.0611492104; Liang WC et al. J Biol Chem. 2006 Jan 13;281(2):951-61. doi: 10.1074/jbc.M508199200)

Lines: 83-92: Literature including reviews (e. g. Annese et al.
Cancers 2019 Mar 18;11(3):381. doi: 10.3390/cancers11030381; Grobbelaar et al. Curr Cancer Drug Targets) 2024 Jan 31.

 doi: 10.2174/0115680096284588240105051402. Online ahead of print) contradict the statement that the role of angiogenesis in pancreatic cancer is unknown and concentrates on one molecule (and its receptor).

Lines 225-226: How can tumor cells hijack the host immune machinery in a immunodeficient mouse? Please characterize the dko mouse used with regard to remaining immune cells (e. g. status of macrophages and other innate immune cells mentionned in the text above).

Line 228: ‚mouse cytokine assay optimised‘ suggest specificity to mouse (and not human) analytes – which is at least ‚not proven‘.

Line 231: Please define live necropsy

Line 233: Please define ‚visual metastasis‘ (e. g. by actual size, resolution of imaging system)

Line 233-235: Please check this sentence.

2.3. Biophotonic imaging: Please include procedure for tissue procurement, sample size, equipment (e. g. camera, resolution, wavelength) and image analysis. Is this like ‚low magnification‘ fluorescence microscopy?

2.4. Thank you for the reference to previous O.P. experiments in this model.

Line 284: see comment above on cell ‚clones‘

Line 297: in the final lay-out, make sure it is ‚2‘ in superskript not ‚square‘

Line 306: Please clarify if it is ‚SPF‘ or really ‚sterile‘ conditions the mice were handled.

Line 308: Please indicate the gender of the mice.

2.8. Please crosscheck the analytes with the introduction section.

Line 237: Thank you for adding „Crossreactivity against human cytokines was not tested.“ Please make sure this does not lead to the impresson (in the whole manuscript), that detction of human cytokines is ‚almost excluded/highly unlikely.

Lines 356-370: Please double check the sentences. ‚Primary antibody‘ implies (missing) secondary antibody (contradicted by the conjugation with a fluorochrome).  Please give the ordering number for the antibody.  

Please explain the rational of testing for mouse (=host) CD31 cells in the blood (and not the tumor).

Results

Line 392: ‚better engraftment than any other mouse strain‘ – which other mouse strains have been tested?

Line 394: Was the baseline monitoring after (or before) tumor cell injection?

                      First tumor measurements including ‚solvent bubble‘

Line 397/398: Caveat: Body weight is always a valuable parameter. Body score index has additional parameters, that may be needed when weight loss could be covered by e. g. by tumor induced oedema formation. Sudden weight gain can be a marker for burden.

Lines 399-403: Sentences double the information. Details on injection prcedure belong to M&M section.

Line 400-414 and Fig 1B/C/D: It is unclear, when the tumors were excised, that is how many animals were analysed/included in the graphs at which timepoint. Body weight ends day 65, Kaplan-Meier and Tumor size on day 70. The ‚green‘ line ends on different days – please use the same color coding for treatment groups in all graphs. Please explain the number of animals included in each analysis. Grous sizes are diffenerent in 1B and 1D and not given in 1E – these parameters are ‚paired‘ so the same animal must be included in all 3 graphs. Number of animals/tumors must be given for Fig 1C (including day of analysis and explantation of ‚lost‘ tumors).

Figure 1 A (fotographs of in situ tumors) does not provide additional information. Only 2 groups are shown. Missing scale bars tumor size cannot be evaluated. Usually only  graphs of tumor size over time (Fig 1B) and/or weight of exiced tumors (Fig 1C) are shown.

Figure 1B: Tumor volume for OP200 is much lower at treatment start – besides n=4 is low.

Figure 1D: Usually tumor weight is given from tumor cell injection. Especially in light of the quite diffenent weights/group on treatment start.

How were the animals randomized into groups? By tumor size at treatment start or by cage groups (for group stability and therefore animal welfare)?

Why and how many animals have been ended on d47 post-cell injection (d12 after treatment start)? Tumor volume at treatment start (d35) should be given.

Figure 2:

Figures 2A-C: The experimental setup and the graphic depiction is unclear to this reviewer. 3 treatment regimes are given (number of OP-injection). Yet only 1 treatment line is shown per graph. Timing of injections and explanation for tumor volume on d42 is missing. In addition the growth curves end much later. Growth inhibition should be shown in a bar graph. Since group sizes seem to be small, ‚real‘ tumor sizes (with error bars) should be given instead of normalised values.

Tumor volumes at treatment start seem much smaller than in Figure 1.

Figures 2 D-E: The combination of 3 panels for 4 groups with overlays of tumor size and CD31 counts is very hard to comprehend – especially since the control groups is the same.

Lines 422-424: The ‚dramatic deline of CD31 cells at time of signs of tumor growth‘  is not represented in the graphs (flat tumor curves)

Figures 2G-I Please give an explanation, how the tumors and treatment groups have been selected for the figure (not all groups represented). It seems that the treatment regimen is different from Fig2A-C (8 injections). Photos ‚with fur‘ are not informative. Scaling of the tumors is not identical. From the images presented (in with given size and resolution), differences in vasculature leading to the tumor are hard to evaluate. Tumorangiogenesis is usually defined as (quantification of) vessels inside the tumor.

Figure3: Unfortunately in the pdf available to this reviewer, Fig 3 is not complete (the lower lane fo images is missing partially).

Image sizes of all A,B and C should be adjusted to form lanes.

Scaling is missing. From the photgraphs, it is hard to judge metastasis. Sample (=tissue) areas are not visible, signals for tumors look quite variable. It would be more informative, to give a table with group size, percentage (and number) of organs with metastasis (per organsite) for each group. Since the tumor cells were injected s. c., has the pancreas as ‚primary‘ site been analysed?

Unfortunately the resolution of the H/E stainings in the pdf are too low. Tumors seem to have quite variable stainig which should be explained (e. g. tumor necrosis). If angiogenesis is affected, it may be possible to show this in H/E (since CD31 stainings are missing). For liver histology, usually tumor areas are marked (arrrows or punctated surcumfences).

Line 599: Blood supply and therefore vasculature is necessary for tumorgrowth. This is a complex process, with neoangiogenesis beeing only one possible mechanism. Metastatic spread via blood vessels is the major mechanism, but not the only one (e. g. lymphatic spread).

Figure 4  is duplicated in the pdf provided to this revieser.

Please explain, why blood analysis has stopped on d45 – the time when tumor growth was greatest.

Line571: The statement that tumor size started to increase on d35 is in contrast to the statements for CD31-blood analysis.

Figure 4: Here the cytokine levesl are given in pg/ml, yet in the M&M section it is stated, that cytokines were measures in relation to a ‚positive control‘. Please explain, if or what standards have been used.

Line 574 and Figure 4: It has been stated that blood has been taken every 7 days, yet cytokine levels are given for d0 and d5.

Line 737: Tumor growth starts at day 30. Please check for congruency on tumor growth timing in the manuscript.

Interestingly the authors refer to the analytes tested only with reference to immune cells (citing reviews), but these discussed immune cells are not present in the mouse strain used (as the authors correctly state several times).

It is unclear to this reviever, what indications the authors have for the stated timeline of tumor vasculature formation. It is generally agreed to, that hypoxia inside the induced by its growth (distance to existing blood supply). Therefore an explanation (and data) would be necessary for angiogenesis to start before tumor growth (and stop while the tumor ‚explodes‘ in size).

The fact that treatment with OP reduces tumor size, is no proof per se that Neu-1 is causing angiogenesis. Smaller tumors could be the cause of less angiogenesis.

Figure 5: Unfortunately the resolution of the Figure is too low, too read the axis scaling. Yet the cytokine level (e. g. for VEGF) seem to bei quite different before d35 (treatment start). Either the scalings of the axis make comparisions difficult/misleading, or a substantial problem exists for interpretation of OP treatment, because of high variability a treatment start.

Starting in the results sections is a discussion of pathways and especially receptors – all not analysed in any of the presented data. Therefore this is speculation and belongs to the discussion section. Nevertheless this reviewer asks the authors keep caution in data presentation and their interpretation.

Line 918: This is not the first manuscript to look at tumor microenvironement (and angiogenesis) in human pancreas tumors grafted into mice. In addition, while sc tumors are easier to monitor, this model has limitations in especially TME. Orthotopic tumors contain per definition a more realistic TME.

In summary, this revised, resubmitted version of the manuscript as improved, but still has major issues in reporting the experiments and especially in overinterpreting the results. Results of blood analysis contribute to the understanding and would be extremly helpfull in monitoring tumor models in mice (especially othotopic ones). But this manuscript lacks comparing data from tumor tissue (same analytes, time course of angiogenesis), which may be beyond the scope of this manuscript. Data presentation should be revised to make it easy to comprehend for the reader, how the experiments were exactly performed (inclunding variable tumor growth in different experiments), what number of animals were analysed at what time. While putting the cytokines into context of their signalling pathways is important, interpretation of blood cytokine effects should be extremely cautious. Especially since an immunodeficitent mouse has (to be) used and therefore major cellular players are missing.

Comments on the Quality of English Language

The Quality of Englisch language is good. 

Proof reading is nessecary, once the correction modus is removed.

Author Response

REVIEWER #1

 Comments and Suggestions for Authors

The manuscript claims to provide novel mechanisms for pancreatic tumor progression, especially tumorangiogenesis. While the approach of analysis cytokine profiles is interesting, the manuscript has severe flaws.

This manuscripts misses the fact, that systemic analysis (like blood) cannot reflect the TME in full. It solely analyses blood sample. Especially angiogenesis is a local reaction (within the tumor).

Author response: Thank you for the comment. We have revised the title of the manuscript to reflect the data of the study, namely, “Contemporaneous Inflammatory, Angiogenic, Fibrogenic and Angiostatic Cytokine Profiles of the Time-to-Tumor Development and Progression Orchestrated by Cancer Cells to Develop Tumor Neovascularization, Progression and Metastasis.” 

Reviewers response: Thank you for the changes

 It is correct that blood analysis does not reflect the TME fully. In the revised manuscript, we have included new data demonstrating the necropsy tumor bloody neovascularization of the untreated control cohort and the OP-treated cohorts in the new Figures 1-3. In Figure 3, biophotonic images of live necropsy of a—tumor, b—liver, c—lung, d—spleen and e—heart from the untreated cohort, 2mg/kg O.P. cohort, 10mg/kg O.P. cohort, 50mg/kg O.P. cohort, and 100 mg/kg O.P. cohort. We have included in Fig 3(B) representative images of tumor neovascularization from untreated cohort and OP-treated cohorts as well as in Fig 3(C) the H&E staining of tumor and liver from untreated cohort and OP-treated cohorts.

The data in Figure 3A reveal the biophotonic images of MiaPaCa-2-eGFP tumors and their migration to the liver, lung, spleen and heart, revealing minimal or negligible metastatic MiaPaCa-2-eGFP of OP-treated cohorts compared to the untreated control. Notably, the tumors from the OP-treated cohorts show fewer bloody vessels/u compared to the untreated cohort, confirming the O.P. affecting the tumor neovascularization process. 

Reviewer's response:  Blood analyses can only show a part of tumor-induced changes. Commonly for TME tumor tissue and for immunological changes spleen and lymph nodes are analysed. Results often show differential profiles depending on tissue analysed. Nevertheless changes in blood parameters could be a very important – as surrogate marker for tumor progression and treatment success. Therefore the now included tumor data are essential for this manuscript.   Macroscopic tumor photographs are a starting point for tumor angiogenesis – but the common marker is CD31 immunohistochemistry. Metastasis: Only one tumor cell line has been used in one (highly) immunodeficient mouse strain. Only blood analysis (cytokines and one cell type) have been performed. Given that narrow spectrum, the authors should use more caution in general claims

Author response: Thank you for the comment. The present study was an extensive analyses of the host cytokine profiles based on previously peer-reviewed published data (Gilmour et al. Cellular Signalling 25 (2013) 2587–2603). That article provided the proof-of-evidence for therapeutic targeting of Neu1 with O.P. in impeding human pancreatic cancer growth and metastatic spread in heterotopic xenografts of ueGFP-MiaPaCa-2 tumors/u growing in RAGxCγ double mutant mice.  

Reviewer's response: There seems to be a misunderstanding of this reviewer’s comments. The validity of the s. c. MiaPaCa-2 Xenograft model has not been questioned, nor strategy of using O.P.  The comments only referred to making very general claims on (pancreatic) cancer and more or less neglecting the systemic limitations of xenografting. The present study analyzed the host angiogenic, fibrogenic and pro-inflammatory cytokine profiling in the tumor-bearing mice by significantly reducing host angiostatic and pro-inflammatory cytokines that restrain tumor development and increasing those for tumor growth. The data suggest an entirely new regulation by cancer cells for tumor development. The findings identify for the first time how pancreatic cancer cells use host cytokine profiling to orchestrate the initiation and tumor development. Necropsy tumor images reveal the extent of the bloody vasculature of the tumor in new Figures 1-3. 

 Reviewer's response: To this reviewer, it is still not clear why blood analysis would detect ‘host cytokines’. It could very well originate from tumor cells.   Here are mso-fareast-font-family:  MsoListParagraphCxSpFirst align

  1. MiaPaca-2 cells express at IL-18 (but not IL-1beta, IL-10) in vitro (Bellone et al. Cancer Immunol Immunother. 2006 Jun;55(6):684-98. doi: 10.1007/s00262-005-0047-0).

Author response: Bellone et al. Cancer Immunol Immunother. 2006 Jun;55(6):684-98. “Cytokines were measured in supernatants and sera (from patients and controls) by ELISA. All cell lines expressed IL-8, IL-18, TGF-b1, TGF-b2 and TGF-b3, but not IFN-c and IL-2 transcripts. Expression of IL-1b, IL-6, IL-10, IL-11, IL-13 and IL-12 mRNA was variable. All the above cytokines were detected as soluble proteins in supernatants, except IL-13. Tumor tissues overexpressed IL-1b, IL-6, IL-8, IL-10, IL-11, IL-12 p40, IL-18, IFN-c, TGF-b1, TGF-b2 and TGF-b3 at the mRNA level and IL-1b, IL-18, TGF-b2 and TGF-b3 also at the protein level. One major finding in the study is that pancreatic carcinoma cells produce IL-12 and IL-18, and that higher serum levels of both cytokines were detected in patients than were found in normal subjects.”

Also, another report has shown that the predicted human IL-18 amino acid sequence is 65% homologous over that of the murine IL-18. From these predicted amino acid sequences, it is clear that IL-18 lacks usual leader sequence necessary for the secretion of the mature form of IL-18 across the cell membrane, but instead contains an unusual leader sequence consisting of 35 amino acids at its N-terminus. IL-18 is synthesized as a 24-kDa precursor protein, which is then enzymatically cleaved into an 18-kDa biologically active mature protein by the action of the intracellular cysteine proteinase, IL-1β converting enzyme (also called ICE or caspase-1; Gu et al., 1997). In addition, proteinase 3, a serine protease stored in the granules of neutrophils and monocytes, has been reported to be an alternative extracellular processing enzyme for IL-1β and IL-18 (Fantuzzi and Dinarello, 1999). The immunological microenvironment of pancreatic carcinoma was found to be clearly in an immunosuppressive state, as convincingly illustrated by the aberrant concomitant expression of potent anti-inflammatory cytokines, such as TGF-b and IL-10 and potentially inactive proinflammatory cytokines, such as IL-12 and IL-18 (Cancer Immunol Immunother (2006) 55: 684–698 DOI 10.1007/s00262-005-0047-0).

  1. VEGF (e. g. Gerber et al. Proc Natl Acad Sci U S A. 2007 Feb 27;104(9):3478-83. doi: 10.1073/pnas.0611492104; Liang WC, Wu X, Peale FV, Lee CV, Meng YG, Gutierrez J, Fu L, Malik AK, Gerber HP, Ferrara N, Fuh G.J Biol Chem. 2006 Jan 13;281(2):951-61. doi: 10.1074/jbc.M508199200) has been shown to be expressed by tumor cells.
  2. Il-2 and IFNgamma are not detectable upon antigen-stimulation in BALB/c rag2 cgc dko mice (generated by Zhao et al. Front Genet. 2019 Apr 30:10:401. doi: 10.3389/fgene.2019.00401)

Author response:  Mature mouse IL-2 shares 56% aa sequence identity with human IL-2. It shows strain-specific heterogeneity in an N-terminal region that contains a poly-glutamine stretch. Mouse and human IL-2 exhibit cross-species activity. The receptor for IL-2 consists of three subunits that are present on the cell surface in varying preformed complexes. The 55 kDa IL-2 R alpha is specific for IL-2 and binds with low affinity. The 75 kDa IL-2 R beta, which is also a component of the IL-15 receptor, binds IL-2 with intermediate affinity. The 64 kDa common gamma chain gamma c/IL-2 R gamma, which is shared with the receptors for IL-4, -7, -9, -15, and -21, does not independently interact with IL-2. Upon ligand binding, signal transduction is performed by both IL-2 R beta and gamma c. It drives resting T cells to proliferate and induces IL-2 and IL-2 R alpha synthesis. It contributes to T cell homeostasis by promoting the Fas-induced death of naïve CD4+ T cells but not activated CD4+ memory lymphocytes. IL-2 plays a central role in the expansion and maintenance of regulatory T cells, although it inhibits the development of Th17 polarized cells.

 Therefore the conclusions of changes in host-derived cytokine profliles are still not valid. This does explicitly not negate systemic cytokine changes caused by tumor and treatment (regardless of which cells are producing them).

For cytokine analyses, ‚murine‘ panels have been used. Therefore host has been claimed as source of these cytokines. It should be stated, if crossreactivity against human cytokines has been tested.

Author response: Thank you for the comment. We have started this cross-reactivity not done in M&M 2.7. Luminex Bio-plex microarray mouse cytokine assay. In addition, the company has indicated in the protocol that the assays for each target analyte are screened against all target analytes to confirm low antibody cross-reactivity.

Reviewers response: Thank you for this addition. Crossreactivity as an issue has been stated independently e. g. in Gerber et al. (Int J Mol Sci. 2018 Dec 1;19(12):3836. doi: 10.3390/ijms19123836.). This states a systematic problem in distinguishing mouse and human cytokines and therefore even in a human to mouse xenograft, conclusions about host vs tumor origin must be very carefully made.Unfortunatley no details on the Bio-plex microarray (e. g. manufacturer, ordering number) have been given by the authors. But in the past Luminex panels may have been reactive to both species – at least there are informations that VEGF and PDGF have.

Author response: Luminex Bio-plex microarray mouse cytokine assay:   This assay is calibrated against highly purified recombinant mouse biomarkers. In addition, the company has indicated in the protocol that the assays for each target analyte are screened against all target analytes to confirm low antibody cross-reactivity.

 (Immuno-)Histological analyses or any other approaches for the tumor are completely missing.

Author response: Thank you for the comment.  In the revised manuscript, we have included new data demonstrating the necropsy tumor bloody neovascularization of the untreated control cohort and the OP-treated cohorts in the new Figures 1-3. In Figure 3, biophotonic images of live necropsy of a—tumor, b—liver, c—lung, d—spleen and e—heart from the untreated cohort, 2mg/kg O.P. cohort, 10mg/kg O.P. cohort, 50mg/kg O.P. cohort, and 100 mg/kg O.P. cohort. We have included in Fig 3(B) representative images of tumor neovascularization from untreated cohort and OP-treated cohorts as well as in Fig 3(C) the H&E staining of tumor and liver from untreated cohort and OP-treated cohorts./u The data in Figure 3A reveal the biophotonic Images of MiaPaCa-2-eGFP tumors and their migration to the liver, lung, spleen and heart, revealing minimal or negligible metastatic MiaPaCa-2-eGFP of OP-treated cohorts compared to the untreated control. Notably, the tumors from the OP-treated cohorts show fewer bloody vessels compared to the untreated cohort, confirming the O.P. affecting the tumor neovascularization process. 

Reviewers response: Thank you for adding H/E stainings and photographs of excised tumors. Macroscopic images of tumors present an impression of tumor vasculature – but the common marker for tumor angiogenesis is immunohistology with CD31 and for vessel maturation SMA. If quantification of tumor vasculature is not available, the statements on tumorangiogenesis should be stated more specific e. g. as ‘typical examples of blood vessels leading to the tumor are shown for each treatment group’. The extent of metastasis is not comprehensible by the photographs shown (see comment below).        5e99; mso-themecolor: text2; mso-themetint: 191; mso-fareast-language: EN-CA"The methods section is missing crucial nformationn (e. g. ‚official‘ mouse strain nomenclature, ‚standard‘ mouse housing in formation (dark: light cycle, food and water administration)).

Author response: Thank you for the comment. >  Heterotopic xenograft mouse model of human pancreatic cancer - The xenograft mice used in the study were immunodeficient with double mutations in the recombinase combining activating gene-2 (RAG2) and the common cytokine recep-tor γ chain (Cγ). These RAG2xCγ mice completely lack lymphoid cells (T-cell, B-cell, and NK-cell deficient), show no spontaneous tumor formation, and have normal hematopoietic parameters. The mice were made by inter-crossing and were kept in SPF isolators in the Animal Care Facility, Queen’s University, Kingston, Ontario K7L 3N6, Canada. A colony of mice was established in the facility. Mice were cared for under sterile conditions in micro-isolators or air-filtered cages and were given autoclaved food and water. The mice used were between 6 and 8 weeks of age. Mice used in the study were approved by the Animal Care Committee at Queen’s University. 

Reviewers response: Thank you for the additional information on the mouse strain. Please add the background strain. Even though T-cells are missing, it could be important (e. g. for reproducablitiy) if it is C57B/6 or BALB/c based. Please indicate the gender of the mice used./ Blood samples have been taken weekly for cytokine analyses. When and how much has been taken for cell analysis?

Author response: Thank you for the comment. >  Mouse blood serum collection. Blood was collected retro-orbitally by penetrating the retro-orbital plexus located behind the eye of the mouse. u200 µL of blood was collected with a capillary tube./u Blood was left to coagulate and was then centrifuged. Collected serum was then freshly frozen at  -80oC and analyzed later for cytokine profiling using Luminex Bio-plex microarray mouse cytokine assay. 

Reviewers response: Thank you for the detailed information. For subcutaneaous tumor cell injection, a cell number of 1x10sup6/sup is given with a cell concentration of 0,5-1 x 10sup6/sup cell/ml. That makes 1-2ml per injection – which is an abnormally high injection volume. Resuspending (incl. washing) cells in saline may reduce cell viability 

Author response: Thank you for the comment. This issue is corrected in the revised manuscript. Puromycin-resistance eGFP-transduced MiaPaCa-2 cell clones were grown in a 75 cm² culture flask at 80-90% confluence. The cells were put into solution using TrypLE Ex-press (Gibco) and washed with sterile saline. The cells in solution were centrifuged for 5 min at 900 rpm. The cell pellet was resuspended in sterile normal saline at 6x10sup6/sup cells/mL for 0.2 mL injection of 1.2x10sup6/sup cells subcutaneously implantation into the right back flank of the mouse. Tumor measurements were taken twice a week.

 Reviewers response: Thank you for the detailed information  mso-fareast-font-family:Another remarkable finding is, that tumor growth can only be detected after about 5 weeks. Endpoints for the experiments must be described – expecially in the ‚survival‘ experiment (humane endpoints?!) Author response: Thank you for the comment. We have added the following in M&M: Humane endpoints of the experimental mice were monitored daily by a registered veterinary practitioner. Mice were euthanized when the tumor volume reached 1000 mm3 or a body weight loss of 15%, according to Animal Care Protocol. Tumor measurements were taken twice a week.

 Reviewers response: Thank you for the information on monitoring and endpoints  mso-fareast-font-family:  The formula given for s. c. tumor volume is unusually written. The commonly used depiction for elipsoid tumors is a name

Author response: Thank you for the comment.   Tumor volumes were analyzed using the following formula: Volume = (width square) x length x 1/2. 

Reviewers response: Thank you Selection of eGFP clones of MiaPaCa-2 is mentioned, but not described. In addition it is unclear, if stable clone(s) were established. It is not clear, why eGFP transfected cells and not the parent cell line were used.

Author response: Thank you for the comment. The manuscript clearly described the selection of eGFP clones of MaPaCa-2 cells. We used the eGPF clone MiaPaCa-2 cells to visualize the tumor progression and metastasis using biophotonic imaging/u (Cancer Biology & Genetics Division of the Queen’s Cancer Research Institute). 

 Reviewers response: With the inclusion of biophotonic imaging, the need of using of eGFP cells is clarified.  2.2. eGFP Lentifect purified lentivirus particles  GeneCopoeia Lentifect™ Lentiviral Particles (Cat# LP-EGFP. LV105-0205) are ready-to-use particles. They are produced from a standardized protocol using purified plasmid DNA and the proprietary reagents EndoFectin™ Lenti (for transfection) and Ti-terBoost™ solution. The protocol uses a third-generation self-inactivating packaging system meeting BioSafety Level 2 requirements with the University Biohazard Committee approval. The Lentifect particles include a cytomegalovirus (CMV) promoter for expression of non-tagged eGFP in target cells, and a puromycin resistance marker is used for the sta-bly selection of transduced cells. The ready-to-use lentiviral particles were used for the transduction of MiaPaCa-2 cells.  Briefly, cells were cultured in 6 well tissue culture plates in DMEM medium containing 10% FCS and 5 μg/mL plasmocin. After 24 hrs, after the medium was discarded, 2 mL of 5 μg/mL of media containing polybrene was added to the cells, followed by eGFP lentiviral particles at MOI=6. The plate was mixed and centrifuged at 2500 rpm for 90 min and incubated at 370C in a 5% CO2 humidified incubator for 24 hrs. After washing, the cells were re-cultured in media for an additional 2 days. On day 5, the media containing an optimal 2μg/mL of puromycin, as pre-determined in a cell viability assay, was replaced in the cells. The selection media was added every 40 hrs to expand puromycin-resistance eGFP-transduced MiaPaCa-2 cell clones. uThe transfection efficiency of 90% was determined using fluorescence microscopy (Zeiss Imager M2) and biophotonic imaging (Cancer Biology & Genetics Division of the Queen’s Cancer Research Institute) before implantation into the mice./u 

Reviewers response: Thank you for the detailed description. Based on this information it seems to this reviewer, that a pool of transduced cells – not a selected clone (based on picking single founder cells) were used. mso-fareast-font-family:  Administration of Oseltamirvir is not described sufficiently (carrier/solvent, concentration/volume, administration route, timing of multiple injections).

Author response: Thank you for the comment. We have added the following to clarify this comment: Inhibitors: a name="_Hlk174824162"/aOseltamivir phosphate (O.P.) (La Roche Ltd., Mississauga, Ontario), the active ingredient in Tamiflu, was used at indicated concentrations. To test the efficacy of O.P. at different dosages, mice were I.P. injected with 0.2 mL of O.P. at 2, 10, 50, 100 and 200 mg/kg in sterile phosphate buffered saline (pH 7.4) daily or otherwise indicated starting at day 35 post-implantation. 

Reviewers response: Thank you. / Claims on tumor immune evasion in a pretty much completely immunodeficient mouse seem far-fetched, especially given the fact that (murine) tumors are being comparatively analyzed in differentially immunodeficient/competent mouse strain sets   a name

Author response: Thank you for the comment.   We disagree with this comment. We have included data from biophotonic imaging (Figure 3A). >  The data in Figure 3A reveal the biophotonic images of MiaPaCa-2-eGFP tumors and their migration to the liver, lung, spleen and heart, revealing minimal or negligible metastatic MiaPaCa-2-eGFP of OP-treated cohorts compared to the untreated control. The OP-treated cohorts show minimal tumor neovascularization compared to the untreated cohort (Figure 3B), supporting the biophotonic tumor data in Figure 3A. The present study was an extensive analyses of the host cytokine profiles based on previously peer-reviewed published data (Gilmour et al. Cellular Signalling 25 (2013) 2587–2603). That article provided the proof-of-evidence for therapeutic targeting of Neu1 with O.P. in impeding human pancreatic cancer growth and metastatic spread in heterotopic xenografts of eGFP-MiaPaCa-2 tumors growing in RAGxCγ double mutant mice.   

Reviewers response: This seems to be a misunderstanding. The reviewer’s comment is NOT on the validity of  the model/experiment/treatment including effects on tumor and metastasis, but on claims on ‘immune evasion’ in this immune-deficient setting./ The questions on the methods have to be answered before evaluating the results and discussion sections.

Author response: Thank you for the comment. We have revised the manuscript accordingly mso-fareast-font-family:  Comments on the Quality of English Language. Only minor spelling checks are necessary

Author response: Thank you for the comment. We use Grammarly software to check the quality of writing and plagiarism. The manuscript reveals 6-7% plagiarism, mainly based on the comment wording to describe Materials and Methods.   

Reviewer’s comments on the revised manuscript:    The presented resubmitted manuscript presents interesting data on systemic detection of tumor induces cytokines. These data on blood analysis are of interest for the research community, since commonly tumor tissue and lymphatic tissue is analysed at endpoint or with sophisticated (imaging) methods. Blood analysis allows - even in mice – for a time-course. Therefore finding (or disproving) changes in blood markers for tumor progression and load even in a specific model is of value for the preclincal research community. While for most cytokine and tumor-host interactions, tumor and lymphatic tissue (spleen, lymph nodes) are anlysed and therefore only endpoint data are available, the blood analysis presented in this manuscript have the major advantage of allowing a time-course This reviewer wants to explicitly state, that the comments are not questionning the model, but the extent to which conclusions are drawn. The focus of the reader should be more directed to the data of blood levels in comparison to tumors size, timing and treatment   This reviewer apologizes that some of the comments may be repetitive to the reply of this reviewer to the authors comments of round 1   The authors must take the inherent limitations of the heterotopic xenograft model better into account.    The human MiaPaca-2 cell line has to be implanted into immunodeficient mice. Here a mouse, which lacks the most important tumor fighting immune cells, namely T-, B- and NK cells has been used. This is a well established model – yet no this excludes conclusions towards immune evasion.    Unfortunately the conclusion, that the ‚mouse panels‘ used present host response has to be taken with care, if not to be questioned – without further experiments beyond the scope of this manuscript. The reasons are:  MiaPaca-2 cells express at IL-18 (but not IL-1beta, IL-10) in vitro (Bellone et al. Cancer Immunol Immunother. 2006 Jun;55(6):684-98. doi: 10.1007/s00262-005-0047-0).

  1. VEGF (e. g. Gerber et al. Proc Natl Acad Sci U S A. 2007 Feb 27;104(9):3478-83. doi: 10.1073/pnas.0611492104; Liang WC, Wu X, Peale FV, Lee CV, Meng YG, Gutierrez J, Fu L, Malik AK, Gerber HP, Ferrara N, Fuh G.J Biol Chem. 2006 Jan 13;281(2):951-61. doi: 10.1074/jbc.M508199200) has been shown to be expressed by tumor cells.
  2. Il-2 and IFNgamma are not detectable upon antigen-stimulation in BALB/c rag2 cgc dko mice (generated by Zhao et al. Front Genet. 2019 Apr 30:10:401. doi: 10.3389/fgene.2019.00401)

 Author response: Thank you for the comment.

Information on tumor data has improved. Presentation of tumor size and H/E stainings are important information, but Figures must be improved (see details below). Macroscopic images of tumors present an impression of tumor vasculature – but the common marker for tumor angiogenesis is immunohistology with CD31.

Author response: Thank you for the comment. The study protocol was to monitor the CD31+ cells in the blood during tumor development. It would be immense to study IHC of CD31+ cells in the tumor at each time point in the development. There are four distinct sequential steps in angiogenesis:

(1) degradation of basement membrane by proteases.

(2) migration of endothelial cells (ECs) into the interstitial space and sprouting.

(3) ECs proliferation at the migrating tip.

(4) lumen formation, generation of new basement membrane with the recruitment of pericyte, formation of anastomoses and finally, blood flow.

Our study used fresh blood samples to measure the CD31+ endothelial cells (ECs) during tumor development. The data, for the first time, showed an interesting pattern of CD31+ ECs migration during tumor development.

Also, the article by Mokhtari (below) has shown that when xenografts attain the maximum volume limits quickly, there is a minor degree of vascularity, as observed by immunohistochemical analysis of CD31 and CD34 expression. This would indicate a decline in the migration of CD31+ EC into the interstitial space and sprouting. This hypothesis is exactly what we have observed in CD31+ ECs migration during tumor development.

Mokhtari et al. Cancers 2021, 13, 2784. https://doi.org/10.3390/cancers13112784. 3D Multicellular Stem-Like Human Breast Tumor Spheroids Enhance Tumorigenicity of Orthotopic Xenografts in Athymic Nude Rat Model.

            “Hypoxia, namely, HIF-2α expression, also plays a crucial role in the activation of angiogenesis, providing an additional means of enhancing metastatic potential [21,79,80]. This finding reinforces the importance of the TME in orchestrating the intricate interactions that facilitate tumorigenesis. A varying degree of vascularity was observed in the xenografts, as observed by immunohistochemical analysis of CD31 and CD34 expression. Due to the xenografts attaining the maximum volume limits quickly, a minor degree of vascularity is expected, given that active proliferation was dominant.”

Line 30: ‘mechanisms unknown’: The statement suggests that there is no knowledge on tumorangiogenesis, TME etc. It would be helpful to rephrase that additional information/analysis is still needed. (same for line 38).

Author response: Thank you for the comment. Done

Line 30: ‘TME surrounds tumors’, the term TME generally also includes tumor stroma and immune cells – within the tumor node. It is true, though, that tumor cells are embedded in TME.

Author response: Thank you for the comment. Done

Lines 46-52: It is unclear, how this data prove that host produced cytokines are influenced by MiaPaCa-2 cells. It is true though, that the systemic cytokine profile is changed (= within the host). Injection of tumor cells is a ‘colonisation’ not a tumor ‘initiation’ model. Cytokines including some of the ones included in the panels used in this study have been analysed in pancreatic cancer before (e. g. two references directly relevant to these experiments: Bellone et al. Cancer Immunol Immunother. 2006 Jun;55(6):684-98. doi: 10.1007/s00262-005-0047-0); Gerber et al. Proc Natl Acad Sci U S A. 2007 Feb 27;104(9):3478-83. doi: 10.1073/pnas.0611492104; Liang WC et al. J Biol Chem. 2006 Jan 13;281(2):951-61. doi: 10.1074/jbc.M508199200)

Author response: Thank you for the comment. Revised the statement accordingly - Injection of tumor cells is a ‘colonisation’ not a tumor ‘initiation’ model.

  • Bellone et al. Cancer Immunol Immunother. 2006 Jun;55(6):684-98. “Cytokines were measured in supernatants and sera (from patients and controls) by ELISA. All cell lines expressed IL-8, IL-18, TGF-b1, TGF-b2 and TGF-b3, but not IFN-c and IL-2 transcripts. Expression of IL-1b, IL-6, IL-10, IL-11, IL-13 and IL-12 mRNA was variable. Cytokine detection - Cell free-supernatants from cell lines and venous serum samples, collected from patients prior to surgery and/or chemotherapy (n=65) and, for comparison, from 30 healthy donors (13 m, 17 f, median age 40, range 24–65 years), were assayed for IL-1b, IL-6, IL-8, IL-10, IL-12 total p40, and IL-13, using commercially available ELISA kits from Euroclone (Paignton, Devon, UK), and for IL-18, IL-11, TGF-b1 and TGF-b2 using commercially available ELISA kits from R&D Systems (Abingdon, UK). The minimum detectable cytokine. concentrations were 5 pg/ml, 0.8, 25, 5, 20, 1.5, 12.5, 31.2, 50, and 7 pg/ml, respectively.
  • Gerber et al. Proc Natl Acad Sci U S A. 2007 Feb 27;104(9): contributions of both tumor- and stromal-cell derived VEGF-A to vascularization of human tumors grown in immunodeficient mice hindered direct comparison between the pharmacological effects of anti-VEGF antibodies with different abilities to block host VEGF. Therefore, by gene replacement technology, we engineered mice to express a humanized form of VEGF-A (hum-X VEGF) that is recognized by many anti-VEGF antibodies and has biochemical and biological properties comparable with WT mouse and human VEGF-A. The hum-X VEGF mouse model was then used to compare the activity and safety of a panel of VEGF Mabs with different affinities for VEGF-A.
  • Liang WC et al. J Biol Chem. 2006 Jan 13;281(2):951-61. Cross-species Vascular Endothelial Growth Factor (VEGF)--blocking Antibodies Completely Inhibit the Growth of Human Tumor Xenografts and Measure the Contribution of Stromal VEGF. This model established that blocking VEGF-A is sufficient to fully inhibit tumor  angiogenesis and tumor growth. This conclusion is consistent with recent studies of an endogenously induced pancreatic -cell tumor model where specifically knocking out VEGF expression locally severely inhibited tumorigenesis (10). These observations demonstrated that blocking all sources of VEGF at an early stage of tumorigenesis was sufficient to inhibit tumor progression and thus validated their application for assessing the role of VEGF at different stages of tumorigenesis in a variety of tumor models in mouse.

Lines: 83-92: Literature including reviews (e. g. Annese et al.
Cancers 2019 Mar 18;11(3):381. doi: 10.3390/cancers11030381; Grobbelaar et al. Curr Cancer Drug Targets) 2024 Jan 31. doi: 10.2174/0115680096284588240105051402. Online ahead of print) contradict the statement that the role of angiogenesis in pancreatic cancer is unknown and concentrates on one molecule (and its receptor).

 Author response: Thank you for the comment. We have included the following in the introduction: “Pancreatic tumors do not have extensive vascularization, and thus, the role of angiogenesis for tumor progression in patients with pancreatic cancer is a complex process. For example, Annese et al. [6] reported that pancreatic cancer angiogenesis can be activated by genetic and epigenetic alterations as well as by cellular stromal components of the tumor microenvironment. They proposed that transcription factors should be considered for the development of new antitumor and anti-angiogenic therapeutic approaches. Grobbelaar et al. [7] have reported on the role of angiogenesis regulators promoting disease progression in pancreatic cancer and how these molecules impact resistance to gemcitabine and various therapies against pancreatic cancer. Therefore, research is still needed before a gold treatment standard can be made [8]. Understanding angiogenic drug resistance also highlights the important need for additional research to regulate alternative non-VEGF-related pro-angiogenic pathways.”

Lines 225-226: How can tumor cells hijack the host immune machinery in a immunodeficient mouse? Please characterize the dko mouse used with regard to remaining immune cells (e. g. status of macrophages and other innate immune cells mentionned in the text above).

Author response: Thank you for the comment. We have added the following to the text: “Here, we investigated the mechanism by which early nascent tumor cells hijack the host immune machinery to facilitate tumor vascularization for tumor progression. Here, we used RAG2xgamma c (Cγ) double mutant mice that are completely alymphoid (T-, B-, NK-cells), do not form spontaneous tumors but exhibit normal hematopoietic cellular parameters. The RAG2x Cγ double mutant mice have decreased dendritic cells, macrophage cells and neutrophils and lack functional receptors for IL-2, IL-4, IL-7, IL-9 and IL-15 cytokines.”

Line 228: ‚mouse cytokine assay optimised‘ suggest specificity to mouse (and not human) analytes – which is at least ‚not proven‘.

Author response: Thank you for the comment. We have added in the M&M section on the assay description: “Cross-reactivity against human cytokines was not tested.”

Line 231: Please define live necropsy

Author response: Thank you for the comment. We have added: “Biophotonic images of live, non-processed necropsy” Also, “Biophotonic images of live freshly non-processed necropsy of tumor, liver, lung, spleen and heart from the untreated cohort and treated cohorts  2mg/kg OP, 10mg/kg OP, 50mg/kg OP, and 100 mg/kg OP to reveal migration of MiaPaCa-2-eGFP to these organs to visualize metastasis. The tissues were placed on a petic dish, and the images of tissue fluorescence were analyzed with a low-magnification fluorescent microscope. The enhanced green fluorescent protein (eGFP) has an excitation peak at 488 nm (blue light) and emits light maximally at 507 nm.”

Line 233: Please define ‚visual metastasis‘ (e. g. by actual size, resolution of imaging system)

Author response: Thank you for the comment. We have added “Biophotonic images of live freshly non-processed necropsy of tumor, liver, lung, spleen and heart from the untreated cohort and treated cohorts  2mg/kg OP, 10mg/kg OP, 50mg/kg OP, and 100 mg/kg OP to reveal migration of MiaPaCa-2-eGFP to these organs to visualize metastasis. The tissues were placed on a petic dish, and the images of tissue fluorescence were analyzed with a low-magnification fluorescent microscope. The enhanced green fluorescent protein (eGFP) has an excitation peak at 488 nm (blue light) and emits light maximally at 507 nm.“

Line 233-235: Please check this sentence.

 Author response: Thank you for the comment. We have revised this section (please see above).

2.3. Biophotonic imaging: Please include procedure for tissue procurement, sample size, equipment (e. g. camera, resolution, wavelength) and image analysis. Is this like ‚low magnification‘ fluorescence microscopy?

Author response: Thank you for the comment. We have added “Biophotonic images of live freshly non-processed necropsy of tumor, liver, lung, spleen and heart from the untreated cohort and treated cohorts  2mg/kg OP, 10mg/kg OP, 50mg/kg OP, and 100 mg/kg OP to reveal migration of MiaPaCa-2-eGFP to these organs to visualize metastasis. The tissues were placed on a petic dish, and the images of tissue fluorescence were analyzed with a low-magnification fluorescent microscope. The enhanced green fluorescent protein (eGFP) has an excitation peak at 488 nm (blue light) and emits light maximally at 507 nm.“

2.4. Thank you for the reference to previous O.P. experiments in this model.

 Line 284: see comment above on cell ‚clones‘

Line 297: in the final lay-out, make sure it is ‚2‘ in superskript not ‚square‘

Author response: Thank you for the comment for the above.

Line 306: Please clarify if it is ‚SPF‘ or really ‚sterile‘ conditions the mice were handled.

Author response: Thank you for the comment. We have added the following:The mice were made by inter-crossing and were kept in specific pathogen-free isolators in the Animal Care Facility, Queen’s University, Kingston, Ontario K7L 3N6, Canada. SPF animal research facility provides breeding, housing, and procedural space for animals free of a defined list of pathogens. A colony of mice was established in the facility. Mice were cared for under sterile conditions in micro-isolators or air-filtered cages and were given autoclaved food and water. The female mice used in the study were between 6 and 8 weeks of age. The Animal Care Committee approved mice used in the study, protocol number 2017-1708, at Queen’s University.”

Line 308: Please indicate the gender of the mice.

 Author response: Thank you for the comment. We have added “The female mice used in the study were between 6 and 8 weeks of age.”

2.8. Please crosscheck the analytes with the introduction section.

Author response: Thank you for the comment. We have crosschecked and added  IL-1β: „pro-inflammatory (IL-1β, IL-10, IFN-γ and TNF-α)“

Line 237: Thank you for adding „Crossreactivity against human cytokines was not tested.“ Please make sure this does not lead to the impresson (in the whole manuscript), that detction of human cytokines is ‚almost excluded/highly unlikely.

 Author response: Thank you for the comment. DONE

Lines 356-370: Please double check the sentences. ‚Primary antibody‘ implies (missing) secondary antibody (contradicted by the conjugation with a fluorochrome).  Please give the ordering number for the antibody.  

Author response: Thank you for the comment. DONE

Please explain the rational of testing for mouse (=host) CD31 cells in the blood (and not the tumor).

Author response: Thank you for the comment. The study protocol was to monitor the CD31+ cells in the blood during tumor development. It would be immense to study IHC of CD31+ cells in the tumor at each time point in the development. There are four distinct sequential steps in angiogenesis:

(1) degradation of basement membrane by proteases.

(2) migration of endothelial cells (ECs) into the interstitial space and sprouting.

(3) ECs proliferation at the migrating tip.

(4) lumen formation, generation of new basement membrane with the recruitment of pericyte, formation of anastomoses and finally, blood flow.

Our study used fresh blood samples to measure the CD31+ endothelial cells (ECs) during tumor development. The data, for the first time, showed an interesting pattern of CD31+ ECs migration during tumor development.”

Results

Line 392: ‚better engraftment than any other mouse strain‘ – which other mouse strains have been tested?

Author response: Thank you for the comment. This has been removed.

Line 394: Was the baseline monitoring after (or before) tumor cell injection? First tumor measurements including ‚solvent bubble‘

Author response: Thank you for the comment. before tumor cell injection

Line 397/398: Caveat: Body weight is always a valuable parameter. Body score index has additional parameters, that may be needed when weight loss could be covered by e. g. by tumor induced oedema formation. Sudden weight gain can be a marker for burden.

Author response: Thank you for the comment. The experimental mice were monitored on a daily basis for any adverse side effects including body score as a routine Vet protocol by the Vet technician.When tumor measurement were made, body weight was performed at the same time.

Lines 399-403: Sentences double the information. Details on injection prcedure belong to M&M section.

Author response: Thank you for the comment. I like to describe here the protocol on how the tumors were measured and body scoring for health. This is important to reiterate to set the reader for the experimental design.

Line 400-414 and Fig 1B/C/D: It is unclear, when the tumors were excised, that is how many animals were analysed/included in the graphs at which timepoint. Body weight ends day 65, Kaplan-Meier and Tumor size on day 70. The ‚green‘ line ends on different days – please use the same color coding for treatment groups in all graphs. Please explain the number of animals included in each analysis. Grous sizes are diffenerent in 1B and 1D and not given in 1E – these parameters are ‚paired‘ so the same animal must be included in all 3 graphs. Number of animals/tumors must be given for Fig 1C (including day of analysis and explantation of ‚lost‘ tumors).

 Author response: Thank you for the comment.

  • Body weight ended on day 65, whereas the Kaplan-Meier and Tumor size ended on day 70 when the mice were euthanized. We  did not do body weight on day 70.
  • The ‚green‘ line overlaps with the other color lines especially with the Kaplan-Meier graph.
  • the number of animals included in each analysis are indicated in the graphs as n=5 or 6.
  • Grous sizes are diffenerent in 1B and 1D and not given in 1E??
  • Fig 1C show a dot plot from each mouse tumor weights. Some of the tumor wts were not done.

Figure 1 A (fotographs of in situ tumors) does not provide additional information. Only 2 groups are shown. Missing scale bars tumor size cannot be evaluated. Usually only  graphs of tumor size over time (Fig 1B) and/or weight of exiced tumors (Fig 1C) are shown.

Author response: Thank you for the comment. Fig1A shows a representation of the vasculature of the  necropsy tumors at day 47 for the untreated and 100 mg/kg OP cohort mouse. This is the same representation for Fig 2G-I. Fig1A only shows a representation of the tumor after opening the skin revealing the vasculature.

 Figure 1B: Tumor volume for OP200 is much lower at treatment start – besides n=4 is low.  

Figure 1D: Usually tumor weight is given from tumor cell injection. Especially in light of the quite diffenent weights/group on treatment start.

How were the animals randomized into groups? By tumor size at treatment start or by cage groups (for group stability and therefore animal welfare)?

Why and how many animals have been ended on d47 post-cell injection (d12 after treatment start)? Tumor volume at treatment start (d35) should be given.

Author response: Thank you for the comment. 

  1. The OP 200 is what we measure.
  2. Usually tumor weight is given from tumor cell injection?? Do not understand this
  3. animals randomized by cage
  4. Untreated mice ended at day 47 due to large tumor volume. We took one mouse from the OP100 to see the degree of vascularization of the tumor under the skin.

Figure 2: Figures 2A-C: The experimental setup and the graphic depiction is unclear to this reviewer. 3 treatment regimes are given (number of OP-injection). Yet only 1 treatment line is shown per graph. Timing of injections and explanation for tumor volume on d42 is missing. In addition the growth curves end much later. Growth inhibition should be shown in a bar graph. Since group sizes seem to be small, ‚real‘ tumor sizes (with error bars) should be given instead of normalised values.

Author response: Thank you for the comment. 

  1. Figure 2. (A-C) shows the Tumor growth inhibition rate following one, four and five intraperitoneal injections of oseltamivir phosphate (OP) dosages of 50, 100 and 200 mg/kg starting at day 42 post-implantation of MiaPaCa-2-eGFP cells. Tumor volumes were taken after one injection of OP, then after four and five for the 50, 100 and 20oOP cohorts. The beauty of the experiments is to monitor the tumor growth rate after number of OP injections.
  2. real‘ tumor sizes (with error bars) for the number of mice in each cohort are displayed in the figure.
  3. The percent tumor growth inhibition rate is displayed in the figure. There is no need for a graph.

Tumor volumes at treatment start seem much smaller than in Figure 1.

Author response: Thank you for the comment. Figure 2 is an additive experiment to measure this tumor growth rate following OP injections.

Figures 2 D-E: The combination of 3 panels for 4 groups with overlays of tumor size and CD31 counts is very hard to comprehend – especially since the control groups is the same.

Author response: Thank you for the comment. The figures are very clear.

Lines 422-424: The ‚dramatic deline of CD31 cells at time of signs of tumor growth‘  is not represented in the graphs (flat tumor curves)

Author response: Thank you for the comment. We revised it to “The CD31 staining results indicated an increase in CD31+ blood mouse cells following tumor cell implantation, with a dramatic decline to a flat line at day 20 post-implantation and remain flat at the time of initial signs of tumor growth.”  

Figures 2G-I Please give an explanation, how the tumors and treatment groups have been selected for the figure (not all groups represented). It seems that the treatment regimen is different from Fig2A-C (8 injections). Photos ‚with fur‘ are not informative. Scaling of the tumors is not identical. From the images presented (in with given size and resolution), differences in vasculature leading to the tumor are hard to evaluate. Tumor angiogenesis is usually defined as (quantification of) vessels inside the tumor.

 Author response: Thank you for the comment. Figure 2A-C is an additive experiment to measure this tumor growth rate following OP injections. Figure 2 D-F is another additive experiment.

Figure3: Unfortunately in the pdf available to this reviewer, Fig 3 is not complete (the lower lane fo images is missing partially).

Author response: Thank you for the comment. The editorial team made tracking on a clean uploaded manuscript. I apologize for the view.

Image sizes of all A,B and C should be adjusted to form lanes.

Author response: Thank you for the comment. Do not understand this??

Scaling is missing. From the photgraphs, it is hard to judge metastasis. Sample (=tissue) areas are not visible, signals for tumors look quite variable. It would be more informative, to give a table with group size, percentage (and number) of organs with metastasis (per organsite) for each group. Since the tumor cells were injected s. c., has the pancreas as ‚primary‘ site been analysed?

Author response: Thank you for the comment. Figure 3 shows the necropsy tissues at time of euthanasia and examined with Biophotonic fluorescent microscope. Each tissue was labelled with a letter.

Unfortunately the resolution of the H/E stainings in the pdf are too low. Tumors seem to have quite variable stainig which should be explained (e. g. tumor necrosis). If angiogenesis is affected, it may be possible to show this in H/E (since CD31 stainings are missing). For liver histology, usually tumor areas are marked (arrrows or punctated surcumfences).

 Author response: Thank you for the comment. H&E staining of tumor and liver from untreated cohort and OP-treated cohorts are representative of these cohorts. We do not have any of the tissues to examine CD31+ cells in the tumors.

Line 599: Blood supply and therefore vasculature is necessary for tumorgrowth. This is a complex process, with neoangiogenesis beeing only one possible mechanism. Metastatic spread via blood vessels is the major mechanism, but not the only one (e. g. lymphatic spread).

Author response: Thank you for the comment. We added this to the text.

Figure 4  is duplicated in the pdf provided to this revieser. Please explain, why blood analysis has stopped on d45 – the time when tumor growth was greatest.

Author response: Thank you for the comment. Editorial tracking messed it up.  The cytokine profiles during tumor growth showed the effect .

Line571: The statement that tumor size started to increase on d35 is in contrast to the statements for CD31-blood analysis.

Author response: Thank you for the comment. Two separate experiments 

Figure 4: Here the cytokine levesl are given in pg/ml, yet in the M&M section it is stated, that cytokines were measures in relation to a ‚positive control‘. Please explain, if or what standards have been used.

Author response: Thank you for the comment. positive control standards for each of the cytokine came with the purchased kit.

Line 574 and Figure 4: It has been stated that blood has been taken every 7 days, yet cytokine levels are given for d0 and d5.

 Author response: Thank you for the comment. This is corrected every 5 days.

Line 737: Tumor growth starts at day 30. Please check for congruency on tumor growth timing in the manuscript.

Author response: Thank you for the comment.  DONE

Interestingly the authors refer to the analytes tested only with reference to immune cells (citing reviews), but these discussed immune cells are not present in the mouse strain used (as the authors correctly state several times).

Author response: Thank you for the comment. We used literature citations to describe these cytokines. Our study mouse model has other innate immune cells to produce the cytokines.

It is unclear to this reviever, what indications the authors have for the stated timeline of tumor vasculature formation. It is generally agreed to, that hypoxia inside the induced by its growth (distance to existing blood supply). Therefore an explanation (and data) would be necessary for angiogenesis to start before tumor growth (and stop while the tumor ‚explodes‘ in size).

Author response: Thank you for the comment. We add this explanation in the first paragraph of the discussion.

The fact that treatment with OP reduces tumor size, is no proof per se that Neu-1 is causing angiogenesis. Smaller tumors could be the cause of less angiogenesis.

Author response: Thank you for the comment.  We have published other peer-reviewed articles stating that OP may have an effect on angiogenesis but no known mechism of action.

Figure 5: Unfortunately the resolution of the Figure is too low, too read the axis scaling. Yet the cytokine level (e. g. for VEGF) seem to bei quite different before d35 (treatment start). Either the scalings of the axis make comparisions difficult/misleading, or a substantial problem exists for interpretation of OP treatment, because of high variability a treatment start.

 Author response: Thank you for the comment.  All figures are imported into the manuscript in high resolution. The editorial tracking made it hard to read.

Starting in the results sections is a discussion of pathways and especially receptors – all not analysed in any of the presented data. Therefore this is speculation and belongs to the discussion section. Nevertheless this reviewer asks the authors keep caution in data presentation and their interpretation.

 Author response: Thank you for the comment.

Line 918: This is not the first manuscript to look at tumor microenvironment (and angiogenesis) in human pancreas tumors grafted into mice. In addition, while sc tumors are easier to monitor, this model has limitations in especially TME. Orthotopic tumors contain per definition a more realistic TME.

 Author response: Thank you for the comment. Agree. Visualization of the vasculature of Orthotopic tumors would involve extensive mouse study. However, this present study provides a foundation to understand the complicity of the system and how to treat it.

In summary, this revised, resubmitted version of the manuscript as improved, but still has major issues in reporting the experiments and especially in overinterpreting the results. Results of blood analysis contribute to the understanding and would be extremly helpfull in monitoring tumor models in mice (especially othotopic ones). But this manuscript lacks comparing data from tumor tissue (same analytes, time course of angiogenesis), which may be beyond the scope of this manuscript. Data presentation should be revised to make it easy to comprehend for the reader, how the experiments were exactly performed (inclunding variable tumor growth in different experiments), what number of animals were analysed at what time. While putting the cytokines into context of their signalling pathways is important, interpretation of blood cytokine effects should be extremely cautious. Especially since an immunodeficitent mouse has (to be) used and therefore major cellular players are missing.

Author response: Thank you for the comment.

 Comments on the Quality of English Language

The Quality of Englisch language is good. 

Proof reading is nessecary, once the correction modus is removed.

Reviewer 2 Report

Comments and Suggestions for Authors

In the context of pancreatic cancer, the use of several methodologies to evaluate various elements of tumor progression and the effects of OP treatment could show the benefits of oseltamivir phosphate.

Reading an article in this format with track changes is really challenging.

The figures and the legends should be arranged.

Expand the influence of the cytokine interactions and how they affect tumor growth collectively.

Could you provide some potential therapeutic applications.

Identify and discuss the study's shortcomings.

Author Response

Author's response to Reviwer-02

In the context of pancreatic cancer, the use of several methodologies to evaluate various elements of tumor progression and the effects of OP treatment could show the benefits of oseltamivir phosphate.

Author response: Thank you for the comment and the benefits of OP treatment.

Reading an article in this format with track changes is really challenging.

Author response: Thank you for the comment. The editorial office and team made the track changes.

The figures and the legends should be arranged.

Author response: Thank you for the comment. Not sure what is suggested here?

Expand the influence of the cytokine interactions and how they affect tumor growth collectively.

Author response: Thank you for the comment. We added additional text with citations:

  • Cancer Immunol Immunother (2006) 55: 684–698. DOI 10.1007/s00262-005-0047-0
  • World J Gastroenterol 2003;9(6):1144-1155

Could you provide some potential therapeutic applications.

Author response: Thank you for the comment.

4.1. Clinical relevance

Understanding the functions of these cytokines and how they are impacted by tumor behavior suggests a potential new tool for therapy targets. Cytokine profiling throughout a patient’s treatment rather than just at the beginning could also provide insight into the progression of the disease and inform decisions on the treatment used. Furthermore, the emphasis placed on OP’s ability to regulate tumor microenvironments by affecting certain cytokines and not others help us better understand its effectiveness as a future therapeutic agent.

4.2. Future directions

These studies were conducted for a maximum of 70 days, and results may change beyond that time point. However, these findings suggest that incorporating OP into treatment protocols of patients with cancer will reduce tumor volume, metastasis, and proliferation and change the microenvironment to be more manageable. This can improve patient outcomes by reducing disease severity and improving outcomes. This may be an excellent alternative therapy for patients who are not finding success with other treatments.

Identify and discuss the study's shortcomings.

Author response: Thank you for the comment. We have not analyzed the tumor H&E for CD31+ cells, which would provide confirmatory data on angiogenesis withing the tumor.

Round 2

Reviewer 1 Report

Comments and Suggestions for Authors

The manuscript and the authors reply still give the impression of involvment of the immune system - which by definition cannot be tested in a solely immundeficient model. This could be misleading to readers not familiar with the model. Tumorimmunologists are used to this mouse-strain as part of a differential (multi-mouse strain) approach to define the role of specific immune cells.

The major issues of differential detection of host (mouse) vs tumor (human) and state of the art angiogenesis analysis remain.

It is clear to this reviewer, that analysis of angiogenesis is a large undertaking - most likely beyond the scope of this manuscript. Therefore this should be reflected in the manuscript.

Reviewer comments have been mainly be adressed in the reply but not in text and figures and therefore data are still hard to comprehend.

For example, in their reply the authors refer to multiple mouse experiments - but this crucial information is (still) not clearly presented in the manuscript.

'Biophotonic imaging': It is still unclear, if sections or whole organs have been imaged and how large the negative tissue area should be (what ist the sampling area).

The experiments with increasing numbers of injections remain unclear (end on which day and variation of tumor size, e. g. by STD/SEM).

If previously published data are an essential complementation to this manuscript, they should be explicitly given and discussed (not only referenced).

This manuscript still needs major reworking.

Comments on the Quality of English Language

Minor proof reading required.